# CREDIBLE SAMPLE ELICITATION BY DEEP LEARNING, FOR DEEP LEARNING

## ABSTRACT

It is important to collect credible training samples $(x, y)$ for building data-intensive learning systems (e.g., a deep learning system). In the literature, there is a line of studies on eliciting distributional information from self-interested agents who hold a relevant information. Asking people to report complex distribution $p(x)$, though theoretically viable, is challenging in practice. This is primarily due to the heavy cognitive loads required for human agents to reason and report this high dimensional information. Consider the example where we are interested in building an image classifier via first collecting a certain category of high-dimensional image data. While classical elicitation results apply to eliciting a complex and generative (and continuous) distribution $p(x)$ for this image data, we are interested in eliciting samples $x_i \sim p(x)$ from agents. This paper introduces a deep learning aided method to incentivize credible sample contributions from selfish and rational agents. The challenge to do so is to design an incentive-compatible score function to score each reported sample to induce truthful reports, instead of an arbitrary or even adversarial one. We show that with accurate estimation of a certain $f$-divergence function we are able to achieve approximate incentive compatibility in eliciting truthful samples. We then present an efficient estimator with theoretical guarantee via studying the variational forms of $f$-divergence function. Our work complements the literature of information elicitation via introducing the problem of *sample elicitation*. We also show a connection between this sample elicitation problem and $f$-GAN, and how this connection can help reconstruct an estimator of the distribution based on collected samples.

## 1 INTRODUCTION

The availability of a large quantity of credible samples is crucial for building high-fidelity machine learning models. This is particularly true for deep learning systems that are data-hungry. Arguably, the most scalable way to collect a large amount of training samples is to crowdsource from a decentralized population of agents who hold relevant sample information. The most popular example is the build of ImageNet (Deng et al., 2009).

The main challenge in eliciting private information is to properly score reported information such that the self-interested agent who holds a private information will be incentivized to report truthfully. At a first look, this problem of eliciting quality data is readily solvable with the seminal solution for eliciting distributional information, called the strictly proper scoring rule (Brier, 1950; Winkler, 1969; Savage, 1971; Matheson & Winkler, 1976; Jose et al., 2006; Gneiting & Raftery, 2007): suppose we are interested in eliciting information about a random vector $X = (X_1, ..., X_{d-1}, Y) \in \Omega \subseteq \mathbb{R}^d$, whose probability density function is denoted by $p$ with distribution $\mathbb{P}$. As the mechanism designer, if we have a sample $x$ drawn from the true distribution $\mathbb{P}$, we can apply strictly proper scoring rules to elicit $p$: the agent who holds $p$ will be scored using $S(p, x)$. $S$ is called strictly proper if it holds for any $p$ and $q$ that $\mathbb{E}_{x \sim \mathbb{P}}[S(p, x)] > \mathbb{E}_{x \sim \mathbb{P}}[S(q, x)]$. The above elicitation approach has two main caveats that limited its application:

- When the outcome space $|\Omega|$ is large and is even possibly infinite, it is practically impossible for any human agents to report such a distribution with reasonable efforts. This partially inspired a line of follow-up works on eliciting property of the distributions, which we will discuss later.

- The mechanism designer may not possess any ground truth samples.

In this work we aim to collect credible samples from self-interested agents via studying the problem of *sample elicitation*. Instead of asking each agent to report the entire distribution $p$, we hope to elicit samples drawn from the distribution $\mathbb{P}$ truthfully. We consider the samples $x_p \sim \mathbb{P}$ and $x_q \sim \mathbb{Q}$. In analogy to strictly proper scoring rules[1], we aim to design a score function $S$ s.t. $\mathbb{E}_{x \sim \mathbb{P}}[S(x_p, x')] > \mathbb{E}_{x \sim \mathbb{P}}[S(x_q, x')]$ for any $q \neq p$, where $x'$ is a reference answer that can be defined using elicited reports. Often, this scoring procedure requires reports from multiple peer agents, and $x'$ is chosen as a function of the reported samples from all other agents (e.g., the average across all the reported $x$s, or a randomly selected $x$). This setting will relax the requirements of high reporting complexity, and has wide applications in collecting training samples for machine learning tasks. Indeed our goal resembles similarity to property elicitation (Lambert et al., 2008; Steinwart et al., 2014; Frongillo & Kash, 2015b), but we emphasize that our aims are different - property elicitation aims to elicit statistical properties of a distribution, while ours focus on eliciting samples drawn from the distributions. In certain scenarios, when agents do not have the complete knowledge or power to compute these properties, our setting enables elicitation of individual sample points.

Our challenge lies in accurately evaluating reported samples. We first observe that the $f$-divergence function between two properly defined distributions of the samples can serve the purpose of incentivizing truthful report of samples. We proceed with using deep learning techniques to solve the score function design problem via a data-driven approach. We then propose a variational approach that enables us to estimate the divergence function efficiently using reported samples, via a variational form of the $f$-divergence function, through a deep neutral network. These estimation results help us establish an approximate incentive compatibility in eliciting truthful samples. It is worth to note that our framework also generalizes to the setting where there is no access to ground truth samples, where we can only rely on reported samples. There we show that our estimation results admit an approximate Bayesian Nash Equilibrium for agents to report truthfully. Furthermore, in our estimation framework, we use a generative adversarial approach to reconstruct the distribution from the elicited samples.

We want to emphasize that the deep learning based estimators considered above are able to handle complex data. And with our deep learning solution, we are further able to provide estimates for the divergence functions used for our scoring mechanisms with provable finite sample complexity. In this paper, we focus on developing theoretical guarantees - other parametric families either can not handle complex data, e.g., it is hard to handle images using kernel methods, or do not have provable guarantees on the sample complexity.

Our contributions are three-folds. (1) We tackle the problem of eliciting complex distribution via proposing a sample elicitation framework. Our deep learning aided solution concept makes it practical to solicit complex sample information from human agents. (2) Our framework covers the case when the mechanism designer has no access to ground truth information, which adds contribution to the peer prediction literature. (3) On the technical side, we develop estimators via deep learning techniques with strong theoretical guarantees. This not only helps us establish approximate incentive-compatibility, but also enables the designer to recover the targeted distribution from elicited samples. Our contribution can therefore be summarized as

> *"eliciting credible training samples by deep learning, for deep learning".*

## 1.1 RELATED WORKS

The most relevant literature to our paper is *strictly proper scoring rules* and *property elicitation*. Scoring rules were developed for eliciting truthful prediction (probability) (Brier, 1950; Winkler, 1969; Savage, 1971; Matheson & Winkler, 1976; Jose et al., 2006; Gneiting & Raftery, 2007). Characterization results for strictly proper scoring rules are given in McCarthy (1956); Savage (1971); Gneiting & Raftery (2007). Property elicitation notices the challenge of eliciting complex distributions (Lambert et al., 2008; Steinwart et al., 2014; Frongillo & Kash, 2015b). For instance, Abernethy & Frongillo (2012) characterize the score functions for eliciting linear properties, and Frongillo & Kash (2015a) study the complexity of eliciting properties. Another line of relevant research is peer prediction, where solutions can help elicit private information when the ground truth verification might be missing (De Alfaro et al., 2016; Gao et al., 2016; Kong et al., 2016;

---

[1]Our specific formulation and goal will be different in details.

Kong & Schoenebeck, 2018; 2019). Our work complements the information elicitation literature via proposing and studying the question of sample elicitation via a variational approach to estimate $f$-divergence functions.

Our work also extends the line of work on divergence estimation. The simplest way to estimate divergence starts with the estimation of density function (Wang et al., 2005; Lee & Park, 2006; Wang et al., 2009; Zhang & Grabchak, 2014; Han et al., 2016). Another method based on the variational form (Donsker & Varadhan, 1975) of the divergence function comes into play (Broniatowski & Keziou, 2004; 2009; Nguyen et al., 2010; Kanamori et al., 2011; Ruderman et al., 2012; Sugiyama et al., 2012), where the estimation of divergence is modeled as the estimation of density ratio between two distributions. The variational form of the divergence function also motivates the well-know Generative Adversarial Network (GAN) (Goodfellow et al., 2014), which learns the distribution by minimizing the Kullback-Leibler divergence. Follow-up works include Nowozin et al. (2016); Arjovsky et al. (2017); Gulrajani et al. (2017); Bellemare et al. (2017), with theoretical analysis in Liu et al. (2017); Arora et al. (2017); Liang (2018); Gao et al. (2019). See also Gao et al. (2017); Bu et al. (2018) for this line of work.

## 1.2 NOTATIONS

For the distribution $\mathbb{P}$, we denote by $\mathbb{P}_n$ the empirical distribution given a set of samples $\{x_i\}_{i=1}^n$ following $\mathbb{P}$, i.e., $\mathbb{P}_n = 1/n \cdot \sum_{i=1}^n \delta_{x_i}$, where $\delta_{x_i}$ is the Dirac measure at $x_i$. We denote by $\|v\|_s = (\sum_{i=1}^d |v^{(i)}|^s)^{1/s}$ the $\ell_s$ norm of the vector $v \in \mathbb{R}^d$ where $1 \leq s < \infty$ and $v^{(i)}$ is the $i$-th entry of $v$. We also denote by $\|v\|_\infty = \max_{1 \leq i \leq d} |v^{(i)}|$ the $\ell_\infty$ norm of $v$. For any real-valued continuous function $f \colon \mathcal{X} \to \mathbb{R}$, we denote by $\|f\|_{L_s(\mathbb{P})} := [\int_{\mathcal{X}} |f(x)|^s \, \mathrm{d}\mathbb{P}]^{1/s}$ the $L_s(\mathbb{P})$ norm of $f$ and $\|f\|_s := [\int_{\mathcal{X}} |f(x)|^s \, \mathrm{d}\mu]^{1/s}$ the $L_s(\mu)$ norm of $f(\cdot)$, where $\mu$ is the Lebesgue measure. Also, we denote by $\|f\|_\infty = \sup_{x \in \mathcal{X}} |f(x)|$ the $L_\infty$ norm of $f(\cdot)$. For any real-valued functions $g(\cdot)$ and $h(\cdot)$ defined on some unbounded subset of the real positive numbers, such that $h(\alpha)$ is strictly positive for all large enough values of $\alpha$, we write $g(\alpha) \lesssim h(\alpha)$ and $g(\alpha) = \mathcal{O}(h(\alpha))$ if $|g(\alpha)| \leq c \cdot h(\alpha)$ for some positive absolute constant $c$ and any $\alpha > \alpha_0$, where $\alpha_0$ is a real number. We denote by $[n]$ the set $\{1, 2, \ldots, n\}$.

## 2 PRELIMINARY

We formulate the question of sample elicitation.

## 2.1 SAMPLE ELICITATION

We consider two scenarios. We start with an easier case where we, as the mechanism designer, have access to a certain number of group truth samples. This is a setting that resembles similarity to the proper scoring rule setting. Then we move to the harder case where the inputs to our mechanism can only be elicited samples from agents.

**Multi-sample elicitation with ground truth samples.** Suppose that the agent holds $n$ samples, with each of them independently drawn from $\mathbb{P}$, i.e., $x_i \sim \mathbb{P}$ [2] for $i \in [n]$. The agent can report each sample arbitrarily, which is denoted as $r_i(x_i) : \Omega \to \Omega$. There are $n$ data $\{x_i^*\}_{i \in [n]}$ independently drawn from the ground truth distribution $\mathbb{Q}$ [3]. We are interested in designing a score function $S(\cdot)$ that takes inputs of each $r_i(\cdot)$ and $\{r_j(x_j), x_j^*\}_{j \in [n]}$: $S(r_i(x_i), \{r_j(x_j), x_j^*\}_{j \in [n]})$ such that if the agent believes that $x^*$ is drawn from the same distribution $x^* \sim \mathbb{P}$, then for any $\{r_j(\cdot)\}_{j \in [n]}$, it holds with probability at least $1 - \delta$ that

$$\sum_{i=1}^n \mathbb{E}_{x, x^* \sim \mathbb{P}} \Big[ S\big(x_i, \{x_j, x_j^*\}_{j \in [n]}\big) \Big] \geq \sum_{i=1}^n \mathbb{E}_{x, x^* \sim \mathbb{P}} \Big[ S\big(r_i(x_i), \{r_j(x_j), x_j^*\}_{j \in [n]}\big) \Big] - n \cdot \epsilon.$$

---

[2]Though we use $x$ to denote the samples we are interested in, $x$ potentially includes both the feature and labels $(x, y)$ as in the context of supervised learning.

[3]The number of ground truth samples can be different from $n$, but we keep them the same for simplicity of presentation. It will mainly affect the terms $\delta$ and $\epsilon$ in our estimations.

We name the above as $(\delta, \epsilon)$-**properness** (per sample) for sample elicitation. When $\delta = \epsilon = 0$, it is reduced to the one that is similar to the properness definition in scoring rule literature (Gneiting & Raftery, 2007). We also shorthand $r_i = r_i(x_i)$ when there is no confusion. Agent believes that her samples are generated from the same distribution as of the ground truth samples, i.e., $\mathbb{P}$ and $\mathbb{Q}$ are same distributions.

**Sample elicitation with peer samples.** Suppose there are $n$ agents each holding a sample $x_i \sim \mathbb{P}_i$, where the distributions $\{\mathbb{P}_i\}_{i \in [n]}$ are not necessarily the same - this models the fact that agents can have subjective biases or local observation biases. This is a more standard peer prediction setting. We denote by their joint distribution as $\mathbb{P} = \mathbb{P}_1 \times \mathbb{P}_2 \times .... \times \mathbb{P}_n$.

Similar to the previous setting, each agent can report her sample arbitrarily, which is denoted as $r_i(x_i) : \Omega \rightarrow \Omega$ for any $i \in [n]$. We are interested in designing and characterizing a score function $S(\cdot)$ that takes inputs of each $r_i(\cdot)$ and $\{r_j(x_j)\}_{j \neq i}$: $S(r_i(x_i), \{r_j(x_j)\}_{j \neq i})$ such that for any $\{r_j(\cdot)\}_{j \in [n]}$, it holds with probability at least $1 - \delta$ that

$$\mathbb{E}_{x \sim \mathbb{P}}\Big[S\big(x_i, \{r_j(x_j) = x_j\}_{j \neq i}\big)\Big] \geq \mathbb{E}_{x \sim \mathbb{P}}\Big[S\big(r(x_i), \{r_j(x_j) = x_j\}_{j \neq i}\big)\Big] - \epsilon.$$

We name the above as $(\delta, \epsilon)$-**Bayesian Nash Equilibrium** (BNE) in truthful elicitation. We only require that agents are all aware of above information structure as common knowledge, but they do not need to form beliefs about details of other agents' sample distributions. Each agent's sample is private to herself.

## 2.2 $f$-DIVERGENCE

It is well known that maximizing the expected proper scores is equivalent to minimizing a corresponding Bregman divergence (Gneiting & Raftery, 2007). More generically, we take the perspective that divergence functions have great potentials to serve as score functions for eliciting samples. We define the $f$-divergence between two distributions $\mathbb{P}$ and $\mathbb{Q}$ with probability density function $p$ and $q$, respectively, as

$$D_f(q\|p) = \int p(x) f\left(\frac{q(x)}{p(x)}\right) \mathrm{d}\mu. \tag{2.1}$$

Here $f(\cdot)$ is a function satisfying certain regularity conditions, which will be specified later. Solving our elicitation problem involves evaluating the $D_f(q\|p)$ successively based on the distributions $\mathbb{P}$ and $\mathbb{Q}$, without knowing the probability density functions $p$ and $q$. Therefore, we have to resolve to a form of $D_f(q\|p)$ which does not involve the analytic forms of $p$ and $q$, but instead sample forms. Following from Fenchel's convex duality, it holds that

$$D_f(q\|p) = \max_{t(\cdot)} \mathbb{E}_{x \sim \mathbb{Q}}[t(x)] - \mathbb{E}_{x \sim \mathbb{P}}[f^\dagger(t(x))], \tag{2.2}$$

where $f^\dagger(\cdot)$ is the Fenchel duality of the function $f(\cdot)$, which is defined as $f^\dagger(u) = \sup_{v \in \mathbb{R}}\{uv - f(v)\}$, and the max is taken over all functions $t(\cdot) \colon \Omega \subset \mathbb{R}^d \rightarrow \mathbb{R}$.

## 3 SAMPLE ELICITATION: A GENERATIVE ADVERSARIAL APPROACH

Recall from (2.2) that $D_f(q\|p)$ admits the following variational form:

$$D_f(q\|p) = \max_{t(\cdot)} \mathbb{E}_{x \sim \mathbb{Q}}[t(x)] - \mathbb{E}_{x \sim \mathbb{P}}[f^\dagger(t(x))]. \tag{3.1}$$

We highlight that via functional derivative, (3.1) is solved by $t^*(x; p, q) = f'(\theta^*(x; p, q))$, where $\theta^*(x; p, q) = q(x)/p(x)$ is the density ratio between $p$ and $q$. Our elicitation builds upon such a variational form (3.1) and the following estimators,

$$\widehat{t}(\cdot; p, q) = \operatorname*{argmin}_{t(\cdot)} \mathbb{E}_{x \sim \mathbb{P}_n}[f^\dagger(t(x))] - \mathbb{E}_{x \sim \mathbb{Q}_n}[t(x)],$$

$$\widehat{D}_f(q\|p) = \mathbb{E}_{x \sim \mathbb{Q}_n}[\widehat{t}(x)] - \mathbb{E}_{x \sim \mathbb{P}_n}[f^\dagger(\widehat{t}(x))].$$

## 3.1 ERROR BOUND AND ASSUMPTIONS

Suppose we have the following error bound for estimating $D_f(q\|p)$: for any probability density functions $p$ and $q$, it holds with probability at least $1 - \delta(n)$ that

$$|\widehat{D}_f(q\|p) - D_f(q\|p)| \leq \epsilon(n), \tag{3.2}$$

where $\delta(n)$ and $\epsilon(n)$ will be specified later in Section 4. To obtain such an error bound, we need the following assumptions.

**Assumption 3.1** (Bounded Density Ratio). The density ratio $\theta^*(x; p, q) = q(x)/p(x)$ is bounded such that $0 < \theta_0 \leq \theta^* \leq \theta_1$ holds for positive absolute constants $\theta_0$ and $\theta_1$.

The above assumption is standard in related literature (Nguyen et al., 2010; Suzuki et al., 2008), which requires that the probability density functions $p$ and $q$ lie on a same support. For simplicity of presentation, we assume that this support is $\Omega \subset \mathbb{R}^d$. We define the $\beta$-Hölder function class on $\Omega$ as follows.

**Definition 3.2** ($\beta$-Hölder Function Class). The $\beta$-Hölder function class with radius $M$ is defined as

$$\mathcal{C}_d^\beta(\Omega, M) = \left\{ t(\cdot) \colon \Omega \subset \mathbb{R}^d \to \mathbb{R} \colon \sum_{\|\alpha\|_1 < \beta} \|\partial^\alpha t\|_\infty + \sum_{\|\alpha\|_1 = \lfloor \beta \rfloor} \sup_{x, y \in \Omega, x \neq y} \frac{|\partial^\alpha t(x) - \partial^\alpha t(y)|}{\|x - y\|_\infty^{\beta - \lfloor \beta \rfloor}} \leq M \right\},$$

where $\partial^\alpha = \partial^{\alpha_1} \cdots \partial^{\alpha_d}$ with $\alpha = (\alpha_1, \ldots, \alpha_d) \in \mathbb{N}^d$.

We assume that the function $t^*(\cdot; p, q)$ is $\beta$-Hölder, which guarantees the smoothness of $t^*(\cdot; p, q)$.

**Assumption 3.3** ($\beta$-Hölder Condition). The function $t^*(\cdot; p, q) \in \mathcal{C}_d^\beta(\Omega, M)$ for some positive absolute constants $M$ and $\beta$, where $\mathcal{C}_d^\beta(\Omega, M)$ is the $\beta$-Hölder function class in Definition 3.2.

In addition, we assume that the following regularity conditions hold for the function $f(\cdot)$ in the definition of $f$-divergence in (2.1).

**Assumption 3.4** (Regularity of Divergence Function). The function $f(\cdot)$ is smooth on $[\theta_0, \theta_1]$ and $f(1) = 0$. Also, it holds that

  (i) $f$ is $\mu_0$-strongly convex on $[\theta_0, \theta_1]$, where $\mu_0$ is a positive absolute constant;
  (ii) $f$ has $L_0$-Lipschitz continuous gradient on $[\theta_0, \theta_1]$, where $L_0$ is a positve absolute constant.

We highlight that we only require that the conditions in Assumption 3.4 hold on the interval $[\theta_0, \theta_1]$, where the absolute constants $\theta_0$ and $\theta_1$ are specified in Assumption 3.1. Thus, Assumption 3.4 is mild and it holds for many commonly used functions in the definition of $f$-divergence. For example, in Kullback-Leibler (KL) divergence, we take $f(u) = -\log u$, which satisfies Assumption 3.4; in Jenson-Shannon divergence, we take $f(u) = u \log u - (u + 1) \log(u + 1)$, which also satisfies Assumption 3.4.

We will show that under Assumptions 3.1, 3.3, and 3.4, the bound (3.2) holds. See Theorem 4.2 in Section 4 for details.

## 3.2 MULTI-SAMPLE ELICITATION WITH GROUND TRUTH SAMPLES

In this section, we focus on multi-sample elicitation with ground truth samples. Under this setting, as a reminder, the agent will report multiple samples. After the agent reported her samples, the mechanism designer obtains a set of ground truth samples $\{x_i^*\}_{i \in [n]} \sim \mathbb{Q}$ to serve the purpose of evaluation. This falls into the standard strictly proper scoring rule setting.

Our mechanism is presented in Algorithm 1.

Algorithm 1 consists of two steps: step 1 is to compute the function $\widehat{t}(\cdot; p, q)$, which enables us, in step 2, to pay agent using a linear-transformed estimated divergence between the reported samples and the true samples. We have the following result.

**Theorem 3.5.** The $f$-scoring mechanism in Algorithm 1 achieves $(2\delta(n), 2b\epsilon(n))$-properness.

---

**Algorithm 1** $f$-scoring mechanism for multiple-sample elicitation with ground truth

1. Compute

$$\widehat{t}(\cdot; p, q) = \operatorname*{argmin}_{t(\cdot)} \mathbb{E}_{x \sim \mathbb{P}_n}[f^\dagger(t(x))] - \mathbb{E}_{x^* \sim \mathbb{Q}_n}[t(x^*)].$$

2. For $i \in [n]$, pay reported sample $r_i$ using

$$S\big(r_i, \{r_j, x_j^*\}_{j=1}^n\big) := a - b\big(\mathbb{E}_{x \sim \mathbb{Q}_n}[\widehat{t}(x; p, q)] - f^\dagger(\widehat{t}(r_i; p, q))\big)$$

for some constants $a, b > 0$.

---

The proof is mainly based on the error bound in estimating $f$-divergence and its non-negativity. Not surprisingly, if the agent believes her samples are generated from the same distribution as the ground truth sample, and that our estimator can well characterize the difference between the two set of samples, she will be incentivized to report truthfully to minimize the difference. We defer the proof to Section B.1.

### 3.3 SINGLE-TASK ELICITATION WITHOUT GROUND TRUTH SAMPLES

The above mechanism in Algorithm 1, while intuitive, has the following two caveats:

- The agent needs to report multiple samples (multi-task/sample elicitation);
- Multiple samples from the ground truth distribution are needed.

To deal with such caveats, we consider the single point elicitation in an elicitation without verification setting. Suppose there are $2n$ agents each holding a sample $x_i \sim \mathbb{P}_i$ [4]. We randomly partition the agents into two groups, and denote the joint distributions for each group's samples as $\mathbb{P}$ and $\mathbb{Q}$ with probability density functions $p$ and $q$ for each of the two groups. Correspondingly, there are a set of $n$ agents for each group, respectively, who are required to report their *single* data point according to two distributions $\mathbb{P}$ and $\mathbb{Q}$, i.e., each of them holds $\{x_i^p\}_{i \in [n]} \sim \mathbb{P}$ and $\{x_i^q\}_{i \in [n]} \sim \mathbb{Q}$. As an interesting note, this is also similar to the setup of a Generative Adversarial Network (GAN), where one distribution corresponds to a generative distribution $x \,|\, y = 1$, and another $x \,|\, y = 0$. This is a connection that we will further explore in Section 5 to recover distributions from elicited samples.

We denote by the joint distribution of $p$ and $q$ as $p \oplus q$ (distribution as $\mathbb{P} \oplus \mathbb{Q}$), and the product of the marginal distribution as $p \times q$ (distribution as $\mathbb{P} \times \mathbb{Q}$). We consider the divergence between the two distributions:

$$D_f(p \oplus q \| p \times q) = \max_{t(\cdot)} \mathbb{E}_{\mathbf{x} \sim \mathbb{P} \oplus \mathbb{Q}}[t(\mathbf{x})] - \mathbb{E}_{\mathbf{x} \sim \mathbb{P} \times \mathbb{Q}}[f^\dagger(t(\mathbf{x}))].$$

Motivated by the connection between mutual information and KL divergence, we define generalized $f$-mutual information in the follows, which characterizes the generic connection between a generalized $f$-mutual information and $f$-divergence.

**Definition 3.6** (Kong & Schoenebeck (2019)). The generalized $f$-mutual information between $p$ and $q$ is defined as

$$I_f(p; q) = D_f(p \oplus q \| p \times q)$$

Further it is shown in Kong & Schoenebeck (2018; 2019) that the data processing inequality for mutual information holds for $I_f(p; q)$ when $f$ is strictly convex. We define the following estimators,

$$\widehat{t}(\cdot; p \oplus q, p \times q) = \operatorname*{argmin}_{t(\cdot)} \mathbb{E}_{\mathbf{x} \sim \mathbb{P}_n \times \mathbb{Q}_n}[f^\dagger(t(\mathbf{x}))] - \mathbb{E}_{\mathbf{x} \sim \mathbb{P}_n \oplus \mathbb{Q}_n}[t(\mathbf{x})],$$

$$\widehat{D}_f(p \oplus q \| p \times q) = \mathbb{E}_{\mathbf{x} \sim \mathbb{P}_n \oplus \mathbb{Q}_n}[\widehat{t}(\mathbf{x}; p \oplus q, p \times q)] - \mathbb{E}_{\mathbf{x} \sim \mathbb{P}_n \times \mathbb{Q}_n}[f^\dagger(\widehat{t}(\mathbf{x}; p \oplus q, p \times q))], \quad (3.3)$$

where $\mathbb{P}_n$ and $\mathbb{Q}_n$ are empirical distributions of the reported samples. We denote $\mathbf{x} \sim \mathbb{P}_n \oplus \mathbb{Q}_n \,|\, r_i$ as the conditional distribution when the first variable is fixed with realization $r_i$. Our mechanism is presented in Algorithm 2.

---

[4]This choice of $2n$ is for the simplicity of presentation.

---

**Algorithm 2** $f$-scoring mechanism for sample elicitation

---

1. Compute $\widehat{t}(\cdot; p \oplus q, p \times q) = \operatorname{argmin}_{t(\cdot)} \mathbb{E}_{\mathbf{x} \sim \mathbb{P}_n \times \mathbb{Q}_n}[f^\dagger(t(\mathbf{x}))] - \mathbb{E}_{\mathbf{x} \sim \mathbb{P}_n \oplus \mathbb{Q}_n}[t(\mathbf{x})]$.
2. Pay each reported sample $r_i$ using:

$$S(r_i, \{r_j\}_{j \neq i}) := a + b\big(\mathbb{E}_{\mathbf{x} \sim \mathbb{P}_n \oplus \mathbb{Q}_n | r_i}[\widehat{t}(\mathbf{x}; p \oplus q, p \times q)] - \mathbb{E}_{\mathbf{x} \sim \mathbb{P}_n \times \mathbb{Q}_n | r_i}[f^\dagger(\widehat{t}(\mathbf{x}; p \oplus q, p \times q))]\big)$$

for some constants $a, b > 0$.

---

Similar to Algorithm 1, the main step in Algorithm 2 is to estimate the $f$-divergence between $\mathbb{P}_n \times \mathbb{Q}_n$ and $\mathbb{P}_n \oplus \mathbb{Q}_n$ using reported samples. Then we pay agents using a linear-transformed form of it. We have the following result.

**Theorem 3.7.** The $f$-scoring mechanism in Algorithm 2 achieves $(2\delta(n), 2b\epsilon(n))$-BNE.

The theorem is proved by error bound in estimating $f$-divergence, a max argument, and the data processing inequality for $f$-mutual information. We defer the proof in Section B.2.

The job left for us is to establish the error bound in estimating the $f$-divergence to obtain $\epsilon(n)$ and $\delta(n)$. Roughly speaking, if we solve the optimization problem (3.3) via deep neural networks with proper structure, it holds that

$$\delta(n) = \exp\{-n^{(d-2\beta)/(2\beta+d)} \log^{14} n\}, \qquad \epsilon(n) = c \cdot n^{-2\beta/(2\beta+d)} \log^7 n,$$

where $c$ is a positive absolute constant. We state and prove this result formally in Section 4.

**Remark 3.8.** (1) When the number of samples grows, it holds that $\delta(n)$ and $\epsilon(n)$ decrease to 0 at least polynomially fast, and our guaranteed approximate incentive-compatibility approaches a strict one. (2) Our method or framework handles arbitrary complex information, where the data can be sampled from high dimensional continuous space. (3) The score function requires no prior knowledge. Instead, we design estimation methods purely based on reported sample data. (4) Our framework also covers the case where the mechanism designer has no access to the ground truth, which adds contribution to the peer prediction literature. So far peer prediction results focused on eliciting simple categorical information. Besides handling complex information structure, our approach can also be viewed as a data-driven mechanism for peer prediction problems.

## 4 ESTIMATION OF $f$-DIVERGENCE

In this section, we introduce an estimator of $f$-divergence and establish the statistical rate of convergence, which characterizes $\epsilon(n)$ and $\delta(n)$. For the simplicity of presentation, in the sequel, we estimate the $f$-divergence $D_f(q\|p)$ between distributions $\mathbb{P}$ and $\mathbb{Q}$ with probability density functions $p$ and $q$, respectively. The rate of convergence of estimating $f$-divergence can be easily extended to that of mutual information.

By Section 3, estimating $f$-divergence between $\mathbb{P}$ and $\mathbb{Q}$ is equivalent to solving the following optimization problem,

$$t^*(\cdot; p, q) = \operatorname*{argmin}_{t(\cdot)} \mathbb{E}_{x \sim \mathbb{P}}[f^\dagger(t(x))] - \mathbb{E}_{x \sim \mathbb{Q}}[t(x)],$$

$$D_f(q\|p) = \mathbb{E}_{x \sim \mathbb{Q}}[t^*(x; p, q)] - \mathbb{E}_{x \sim \mathbb{P}}[f^\dagger(t^*(x; p, q))]. \tag{4.1}$$

In what follows, we propose an estimator of $D_f(q\|p)$. By Assumption 3.3, it suffices to solve (4.1) on the function class $\mathcal{C}_d^\beta(\Omega, M)$. To this end, we approximate solution to (4.1) by the family of deep neural networks.

We now define the family of deep neural networks as follows.

**Definition 4.1.** Given a vector $k = (k_0, \ldots, k_{L+1}) \in \mathbb{N}^{L+2}$, where $k_0 = d$ and $k_{L+1} = 1$, the family of deep neural networks is defined as

$$\Phi(L, k) = \{\varphi(x; W, v) = W_{L+1} \sigma_{v_L} \cdots W_2 \sigma_{v_1} W_1 x : W_j \in \mathbb{R}^{k_j \times k_{j-1}}, v_j \in \mathbb{R}^{k_j}\}.$$

Here we write $\sigma_v(x)$ as $\sigma(x - v)$ for notational convenience, where $\sigma(\cdot)$ is the ReLU activation function.

To avoid overfitting, the sparsity of the deep neural networks is a typical assumption in deep learning literature. In practice, such a sparsity property is achieved through certain techniques, e.g., dropout (Srivastava et al., 2014), or certain network architecture, e.g., convolutional neural network (Krizhevsky et al., 2012). We now define the family of sparse networks as follows,

$$\Phi_M(L, k, s) = \{\varphi(x; W, v) \in \Phi(L, d) \colon \|\varphi\|_\infty \leq M, \ \|W_j\|_\infty \leq 1 \text{ for } j \in [L+1],$$

$$\|v_j\|_\infty \leq 1 \text{ for } j \in [L], \ \sum_{j=1}^{L+1} \|W_j\|_0 + \sum_{j=1}^{L} \|v_j\|_0 \leq s\}, \tag{4.2}$$

where $s$ is the sparsity. In contrast, another approach to avoid overfitting is to control the norm of parameters. See Section A.2 for details.

We now propose the following estimators

$$\widehat{t}(x; p, q) = \underset{t \in \Phi_M(L, k, s)}{\operatorname{argmin}} \ \mathbb{E}_{x \sim \mathbb{P}_n}[f^\dagger(t(x))] - \mathbb{E}_{x \sim \mathbb{Q}_n}[t(x)],$$

$$\widehat{D}_f(q\|p) = \mathbb{E}_{x \sim \mathbb{Q}_n}[\widehat{t}(x; p, q)] - \mathbb{E}_{x \sim \mathbb{P}_n}[f^\dagger(\widehat{t}(x; p, q))]. \tag{4.3}$$

The following theorem characterizes the statistical rate of convergence of the estimators defined in (4.3).

**Theorem 4.2.** Let $L = \mathcal{O}(\log n)$, $s = \mathcal{O}(N \log n)$, and $k = (d, d, \mathcal{O}(dN), \mathcal{O}(dN), \ldots, \mathcal{O}(dN), 1)$ in (4.2), where $N = n^{d/(2\beta+d)}$. Under Assumptions 3.1, 3.3, and 3.4, it holds with probability at least $1 - \exp\{-n^{(d-2\beta)/(2\beta+d)} \log^{14} n\}$ that

$$|D_f(q\|p) - \widehat{D}_f(q\|p)| \lesssim n^{-\frac{2\beta}{2\beta+d}} \log^7 n.$$

We defer the proof of the theorem in Section B.3. By Theorem 4.2, the estimators in (4.3) achieve the optimal nonparametric rate of convergence (Stone, 1982) up to a logarithmic term. By (3.2) and Theorem 4.2, we have

$$\delta(n) = \exp\{-n^{(d-2\beta)/(2\beta+d)} \cdot \log^{14} n\}, \qquad \epsilon(n) = c \cdot n^{-2\beta/(2\beta+d)} \cdot \log^7 n,$$

where $c$ is a positive absolute constant.

## 5 CONNECTION TO GAN AND RECONSTRUCTION OF DISTRIBUTION

After sample elicitation, a natural question to ask is how to learn a representative probability density function from the samples. Denote the probability density function from elicited samples as $p$. Then, learning the probability density function $p$ is to solve for

$$q^* = \underset{q \in \mathcal{Q}}{\operatorname{argmin}} \ D_f(q\|p), \tag{5.1}$$

where $\mathcal{Q}$ is the probability density function space.

To see the connection between (5.1) and the formulation of $f$-GAN (Nowozin et al., 2016), by combining (2.2) and (5.1), we have

$$q^* = \underset{q \in \mathcal{Q}}{\operatorname{argmin}} \max_t \mathbb{E}_{x \sim \mathbb{Q}}[t(x)] - \mathbb{E}_{x \sim \mathbb{P}}[f^\dagger(t(x))],$$

which is the formulation of $f$-GAN. Here the probability density function $q(\cdot)$ is the generator, while the function $t(\cdot)$ is the discriminator.

By the non-negativity of $f$-divergence, $q^* = p$ solves (5.1). We now propose the following estimator

$$\widehat{q} = \underset{q \in \mathcal{Q}}{\operatorname{argmin}} \ \widehat{D}_f(q\|p), \tag{5.2}$$

where $\widehat{D}_f(q\|p)$ is given in (4.3).

We define covering number as follows.

**Definition 5.1** (Covering Number). Let $(V, \| \cdot \|_{L_2})$ be a normed space, and $\mathcal{Q} \subset V$. We say that $\{v_1, \ldots, v_N\}$ is a $\delta$-covering over $\mathcal{Q}$ of size $N$ if $\mathcal{Q} \subset \cup_{i=1}^N B(v_i, \delta)$, where $B(v_i, \delta)$ is the $\delta$-ball centered at $v_i$. The covering number is defined as $N_2(\delta, \mathcal{Q}) = \min\{N : \exists \epsilon\text{-covering over } \mathcal{Q} \text{ of size } N\}$.

We impose the following assumption on the covering number of the probability density function space $\mathcal{Q}$.

**Assumption 5.2.** It holds that $N_2(\delta, \mathcal{Q}) = \mathcal{O}(\exp\{1/\delta^{d/(2\beta)-1}\})$.

Recall that $q^* = p$ is the unique minimizer of the problem (5.1). Therefore, the $f$-divergence $D_f(\widehat{q}\|p)$ characterizes the deviation of $\widehat{q}$ from $p^*$. The following theorem characterizes the error bound of estimating $q^*$ by $\widehat{q}$.

**Theorem 5.3.** Under the same assumptions in Theorem 4.2 and Assumption 5.2, for sufficiently large sample size $n$, it holds with probability at least $1 - 1/n$ that

$$D_f(\widehat{q}\|p) \lesssim n^{-\frac{2\beta}{2\beta+d}} \cdot \log^7 n + \min_{\widetilde{q} \in \mathcal{Q}} D_f(\widetilde{q}\|p). \tag{5.3}$$

We defer the proof of the theorem in Section B.4.

In Theorem 5.3, the first term on the RHS of (5.3) characterizes the generalization error of the estimator in (5.2), while the second term characterizes the approximation error. If the approximation error in (5.3) vanishes, then the estimator $\widehat{q}$ converges to the true density function $q^* = p$ at the optimal nonparametric rate of convergence (Stone, 1982) up to a logarithmic term.

## 6 CONCLUDING REMARKS

In this work, we introduce the problem of sample elicitation as an alternative to eliciting complicated distribution. Our elicitation mechanism leverages the variational form of $f$-divergence functions to achieve accurate estimation of the divergences using samples. We provide theoretical guarantee for both our estimators and the achieved incentive compatibility.

It reminds an interesting problem to find out more "organic" mechanisms for sample elicitation that requires (i) less elicited samples; and (ii) induced strict truthfulness instead of approximated ones.

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

# A    AUXILIARY ANALYSIS

## A.1    AUXILIARY RESULTS ON SPARSITY CONTROL

In this section, we provide some auxiliary results on (4.3). We first state an oracle inequality showing the rate of convergence of $\widehat{t}(x; p, q)$.

**Theorem A.1.** Given $0 < \varepsilon < 1$, for any sample size $n$ satisfies that $n \gtrsim [\gamma + \gamma^{-1} \log(1/\varepsilon)]^2$, under Assumptions 3.1, 3.3, and 3.4, it holds that

$$\|\widehat{t} - t^*\|_{L_2(\mathbb{P})} \lesssim \min_{\widetilde{t} \in \Phi_M(L, k, s)} \|\widetilde{t} - t^*\|_{L_2(\mathbb{P})} + \gamma n^{-1/2} \log n + n^{-1/2}[\sqrt{\log(1/\varepsilon)} + \gamma^{-1} \log(1/\varepsilon)]$$

with probability at least $1 - \varepsilon \cdot \exp(-\gamma^2)$. Here $\gamma = s^{1/2} \log(V^2 L)$ and $V = \prod_{j=0}^{L+1}(k_j + 1)$.

We defer the proof of to Section B.5.

As a by-product, note that $t^*(x; p, q) = f'(\theta^*(x; p, q)) = f'(q(x)/p(x))$, based on the error bound established in Theorem A.1, we obtain the following result.

**Corollary A.2.** Given $0 < \varepsilon < 1$, for the sample size $n \gtrsim [\gamma + \gamma^{-1} \log(1/\varepsilon)]^2$, under Assumptions 3.1, 3.3, and 3.4, it holds with probability at least $1 - \varepsilon \cdot \exp(-\gamma^2)$ that

$$\|\widehat{\theta} - \theta^*\|_{L_2(\mathbb{P})} \lesssim \min_{\widetilde{t} \in \Phi_M(L, k, s)} \|\widetilde{t} - t^*\|_{L_2(\mathbb{P})} + \gamma n^{-1/2} \log n + n^{-1/2}[\sqrt{\log(1/\varepsilon)} + \gamma^{-1} \log(1/\varepsilon)].$$

Here $\gamma = s^{1/2} \log(V^2 L)$ and $V = \prod_{j=0}^{L+1}(k_j + 1)$.

*Proof.* Note that $(f')^{-1} = (f^\dagger)'$ and $f^\dagger$ has Lipschitz continuous gradient with parameter $1/\mu_0$ from Assumption 3.4 and Lemma D.6, we obtain the result from Theorem A.1.  □

## A.2    ERROR BOUND USING NORM CONTROL

In this section, we consider using norm of the parameters (specifically speaking, the norm of $W_j$ and $v_j$ in (4.1)) to control the error bound, which is an alternative of the network model shown in (4.2). We consider the family of $L$-layer neural networks with bounded spectral norm for weight matrices $W = \{W_j \in \mathbb{R}^{k_j \times k_{j-1}}\}_{j=1}^{L+1}$, where $k_0 = d$ and $k_{L+1} = 1$, and vector $v = \{v_j \in \mathbb{R}^{k_j}\}_{j=1}^L$, which is denoted as

$$\Phi_{\text{norm}} = \Phi_{\text{norm}}(L, k, A, B) = \{\varphi(x; W, v) \in \Phi(L, k) : \|v_j\|_2 \le A_j \text{ for all } j \in [L],$$
$$\|W_j\|_2 \le B_j \text{ for all } j \in [L+1]\}, \tag{A.1}$$

where $\sigma_{v_j}(x)$ is short for $\sigma(x - v_j)$ for any $j \in [L]$. We write the following optimization problem,

$$\widehat{t}(x; p, q) = \operatorname*{argmin}_{t \in \Phi_{\text{norm}}} \mathbb{E}_{x \sim \mathbb{P}_n}[f^\dagger(t(x))] - \mathbb{E}_{x \sim \mathbb{Q}_n}[t(x)],$$
$$\widehat{D}_f(q\|p) = \mathbb{E}_{x \sim \mathbb{Q}_n}[\widehat{t}(x; p, q)] - \mathbb{E}_{x \sim \mathbb{P}_n}[f^\dagger(\widehat{t}(x; p, q))]. \tag{A.2}$$

Based on this formulation, we derive the error bound on the estimated $f$-divergence in the following theorem. We only consider the generalization error bound in this setting. Therefore, we assume that the ground truth $t^*(x; p, q) = f'(q(x)/p(x))$ locates within $\Phi_{\text{norm}}$. Before we state the theorem, we first define two parameters for the family of neural networks $\Phi_{\text{norm}}(L, k, A, B)$ as follows

$$\gamma_1 = B \prod_{j=1}^{L+1} B_j \cdot \sqrt{\sum_{j=0}^{L+1} k_j^2}, \qquad \gamma_2 = \frac{L \cdot (\sqrt{\sum_{j=1}^{L+1} k_j^2 B_j^2} + \sum_{j=1}^L A_j)}{\sum_{j=0}^{L+1} k_j^2 \cdot \min_j B_j^2} \cdot \sum_{j=1}^L A_j. \tag{A.3}$$

We proceed to state the theorem.

**Theorem A.3.** We assume that $t^*(x; p, q) \in \Phi_{\text{norm}}$. Then for any $0 < \varepsilon < 1$, with probability at least $1 - \varepsilon$, it holds that

$$|\widehat{D}_f(q\|p) - D_f(q\|p)| \lesssim \gamma_1 \cdot n^{-1/2} \log(\gamma_2 n) + \prod_{j=1}^{L+1} B_j \cdot n^{-1/2} \sqrt{\log(1/\varepsilon)}.$$

Here $\gamma_1$ and $\gamma_2$ are defined in (A.3).

We defer the proof to Section B.6.

The next theorem uses the results in Theorem A.3. Recall that in Section §A.2, we assume that the minimizer $t^*$ to the population version problem (4.1) lies within the norm-controlled family of neural networks $\Phi_{\mathrm{norm}}(L, k, A, B)$.

**Theorem A.4.** Recall that we defined the parameter $\gamma_1$ and $\gamma_2$ of the family of neural networks $\Phi_{\mathrm{norm}}(L, k, A, B)$ in (A.3), the estimated distribution $\widehat{q}$ in (5.2), and the ground truth $q^* = p$. We denote the the covering number of the probability distribution function class $\mathcal{Q}$ as $N_2(\delta, \mathcal{Q})$, then for any $0 < \varepsilon < 1$, with probability at least $1 - \varepsilon$, we have

$$D_f(\widehat{q}\|p) \lesssim b_2(n, \gamma_1, \gamma_2) + \prod_{j=1}^{L+1} B_j \cdot n^{-1/2} \cdot \sqrt{\log(N_2[b_2(n, \gamma_1, \gamma_2), \mathcal{Q}]/\varepsilon)} + \min_{\widetilde{q} \in \mathcal{Q}} D_f(\widetilde{q}\|p),$$

where $b_2(n, \gamma_1, \gamma_2) = \gamma_1 n^{-1/2} \log(\gamma_2 n)$.

We defer the proof to Section B.7.

# B    PROOFS OF THEOREMS

## B.1    PROOF OF THEOREM 3.5

If the player truthfully reports, she will receive the following expected payment per sample $i$: with probability at least $1 - \delta(n)$,

$$
\begin{aligned}
\mathbb{E}[S(r_i, \cdot)] &:= a - b(\mathbb{E}_{x \sim \mathbb{Q}_n}[\widehat{t}(x)] - \mathbb{E}_{x_i \sim \mathbb{P}_n}[f^\dagger(\widehat{t}(x_i))]) \\
&= a - b \cdot \widehat{D}_f(q\|p) \\
&\geq a - b \cdot (D_f(q\|p) + \epsilon(n)) \quad \text{(sample complexity guarantee)} \\
&\geq a - b \cdot (D_f(p\|p) + \epsilon(n)) \quad \text{(agent believes } p = q) \\
&= a - b\epsilon(n)
\end{aligned}
$$

Similarly, any misreporting according to a distribution $\widetilde{p}$ with distribution $\widetilde{\mathbb{P}}$ will lead to the following derivation with probability at least $1 - \delta$

$$
\begin{aligned}
\mathbb{E}[S(r_i, \cdot)] &:= a - b(\mathbb{E}_{x \sim \mathbb{Q}_n}[\widehat{t}(x)] - \mathbb{E}_{x_i \sim \widetilde{\mathbb{P}}_n}[f^\dagger(\widehat{t}(x_i))]) \\
&= a - b \cdot \widehat{D}_f(q\|\widetilde{p}) \\
&\leq a - b \cdot (D_f(p\|\widetilde{p}) - \epsilon(n)) \\
&\leq a + b\epsilon(n) \quad \text{(non-negativity of } D_f)
\end{aligned}
$$

Combining above, and using union bound, leads to $(2\delta(n), 2b\epsilon(n))$-properness.

## B.2    PROOF OF THEOREM 3.7

Consider an arbitrary agent $i$. Suppose every other agent truthfully reports.

$$
\begin{aligned}
\mathbb{E}[S(r_i, \{r_j\}_{j \neq i})] &= a + b(\mathbb{E}_{\mathbf{x} \sim \mathbb{P}_n \oplus \mathbb{Q}_n | r_i}[\widehat{t}(\mathbf{x})] - \mathbb{E}_{\mathbf{x} \sim \mathbb{P}_n \times \mathbb{Q}_n | r_i}\{f^\dagger(\widehat{t}(\mathbf{x}))\}) \\
&= a + b\mathbb{E}[\mathbb{E}_{\mathbf{x} \sim \mathbb{P}_n \oplus \mathbb{Q}_n | r_i}[\widehat{t}(x)] - \mathbb{E}_{\mathbf{x} \sim \mathbb{P}_n \times \mathbb{Q}_n | r_i}\{f^\dagger(\widehat{t}(\mathbf{x}))\}]
\end{aligned}
$$

Consider the divergence term $\mathbb{E}[\mathbb{E}_{\mathbf{x}\sim\mathbb{P}_n\oplus\mathbb{Q}_n|r_i}[\widehat{t}(x)] - \mathbb{E}_{\mathbf{x}\sim\mathbb{P}_n\times\mathbb{Q}_n|r_i}\{f^\dagger(\widehat{t}(\mathbf{x}))\}]$. Reporting a $r_i \sim \widetilde{\mathbb{P}} \neq \mathbb{P}$ (denoting its distribution as $\widetilde{p}$) leads to the following score

$$\mathbb{E}_{r_i\sim\widetilde{\mathbb{P}}_n}[\mathbb{E}_{\mathbf{x}\sim\widetilde{\mathbb{P}}_n\oplus\mathbb{Q}_n|r_i}[\widehat{t}(\mathbf{x})] - \mathbb{E}_{\mathbf{x}\sim\widetilde{\mathbb{P}}_n\times\mathbb{Q}_n|r_i}\{f^\dagger(\widehat{t}(\mathbf{x}))\}]$$

$$= \mathbb{E}_{\mathbf{x}\sim\widetilde{\mathbb{P}}_n\oplus\mathbb{Q}_n}[\widehat{t}(\mathbf{x})] - \mathbb{E}_{\mathbf{x}\sim\widetilde{\mathbb{P}}_n\times\mathbb{Q}_n}\{f^\dagger(\widehat{t}(\mathbf{x}))\} \quad \text{(tower property)}$$

$$\leq \max_t \mathbb{E}_{\mathbf{x}\sim\widetilde{\mathbb{P}}_n\oplus\mathbb{Q}_n}[t(\mathbf{x})] - \mathbb{E}_{\mathbf{x}\sim\widetilde{\mathbb{P}}_n\times\mathbb{Q}_n}\{f^\dagger(t(\mathbf{x}))\} \quad \text{(max)}$$

$$= \widehat{D}_f(\widetilde{p}\oplus q\|\widetilde{p}\times q)$$

$$\leq D_f(\widetilde{p}\oplus q\|\widetilde{p}\times q) + \epsilon(n)$$

$$= I_f(\widetilde{p};q) + \epsilon(n) \quad \text{(definition)}$$

$$\leq I_f(p;q) + \epsilon(n) \quad \text{(data processing inequality (Kong \& Schoenebeck, 2019))}$$

with probability at least $1 - \delta(n)$ (the other $\delta(n)$ probability with maximum score $\bar{S}$).

Now we prove that truthful reporting leads at least

$$I_f(p;q) - \epsilon(n)$$

of the divergence term:

$$\mathbb{E}_{x_i\sim\mathbb{P}_n}[\mathbb{E}_{\mathbf{x}\sim\mathbb{P}_n\oplus\mathbb{Q}_n|x_i}[\widehat{t}(\mathbf{x})] - \mathbb{E}_{\mathbf{x}\sim\mathbb{P}_n\times\mathbb{Q}_n|x_i}\{f^\dagger(\widehat{t}(\mathbf{x}))\}]$$

$$= \mathbb{E}_{\mathbf{x}\sim\mathbb{P}_n\oplus\mathbb{Q}_n}[\widehat{t}(\mathbf{x})] - \mathbb{E}_{\mathbf{x}\sim\mathbb{P}_n\times\mathbb{Q}_n}\{f^\dagger(\widehat{t}(\mathbf{x}))\} \quad \text{(tower property)}$$

$$= \widehat{D}_f(p\oplus q\|p\times q)$$

$$\geq D_f(p\oplus q\|p\times q) - \epsilon(n)$$

$$= I_f(p;q) - \epsilon(n) \quad \text{(definition)}$$

with probability at least $1 - \delta(n)$ (the other $\delta(n)$ probability with score at least 0). Therefore the expected divergence terms differ at most by $2\epsilon(n)$ with probability at least $1 - 2\delta(n)$ (via union bound). The above combines to establish a $(2\delta(n), 2b\epsilon(n))$-BNE.

### B.3 PROOF OF THEOREM 4.2

**Step 1.** We proceed to bound $\|t^* - \widehat{t}\|_{L_2(\mathbb{P})}$. We first proceed to find some $\widetilde{t} \in \Phi_M(L, k, s)$. Note that the ground truth $t^*$ lies on a finite support $\Omega \subset [a, b]^d$. To invoke Theorem D.5, we denote $t'(y) = t^*((b - a)y + a\mathbf{1}_d)$, where $\mathbf{1}_d = (1, 1, \ldots, 1)^\top \in \mathbb{R}^d$. Then the support of $t'$ lies in the unit cube $[0, 1]^d$. We choose $L' = \mathcal{O}(\log n)$, $s' = \mathcal{O}(N \log n)$, $k' = (d, \mathcal{O}(dN), \mathcal{O}(dN), \ldots, \mathcal{O}(dN), 1)$, and $m' = \log n$, we then utilize Theorem D.5 to construct some $\widetilde{t'} \in \Phi_M(L', k', s')$ such that

$$\|\widetilde{t'} - t'\|_{L^\infty([0,1]^d)} \lesssim N^{-\beta/d}.$$

We further define $\widetilde{t}(\cdot) = \widetilde{t'} \circ \ell(\cdot)$, where $\ell(\cdot)$ is a linear mapping taking the following form

$$\ell(x) = \frac{x}{b - a} - \frac{a}{b - a} \cdot \mathbf{1}_d.$$

To this end, we know that $\widetilde{t} \in \Phi_M(L, k, s)$, with parameters $L$, $k$, and $s$ given in the statement of Theorem 4.2. We fix this $\widetilde{t}$ and invoke Theorem A.1, then with probability at least $1 - \varepsilon \cdot \exp(-\gamma^2)$, we have

$$\|\widehat{t} - t^*\|_{L_2(\mathbb{P})} \lesssim \|\widetilde{t} - t^*\|_{L_2(\mathbb{P})} + \gamma n^{-1/2}\log n + n^{-1/2}[\sqrt{\log(1/\varepsilon)} + \gamma^{-1}\log(1/\varepsilon)]$$

$$\lesssim N^{-\beta/d} + \gamma n^{-1/2}\log n + n^{-1/2}[\sqrt{\log(1/\varepsilon)} + \gamma^{-1}\log(1/\varepsilon)]. \tag{B.1}$$

Note that $\gamma$ takes the form $\gamma = s^{1/2}\log(V^2 L)$, where $V = \mathcal{O}(d^L \cdot N^L)$ and $L, s$ given in the statement of Theorem 4.2, it holds that $\gamma = \mathcal{O}(N^{1/2}\log^{5/2} n)$. Moreover, by the choice $N = n^{d/(2\beta+d)}$, combining (B.1) and taking $\varepsilon = 1/n$, we know that

$$\|\widehat{t} - t^*\|_{L_2(\mathbb{P})} \lesssim n^{-\beta/(2\beta+d)}\log^{7/2} n \tag{B.2}$$

with probability at least $1 - \exp\{-n^{d/(2\beta+d)} \log^5 n\}$.

**Step 2.** We denote by $\mathcal{L}(t) = \mathbb{E}_{x\sim\mathbb{Q}}[t(x)] - \mathbb{E}_{x\sim\mathbb{P}}[f^\dagger(t(x))]$ and $\widehat{\mathcal{L}}(t) = \mathbb{E}_{x\sim\mathbb{Q}_n}[t(x)] - \mathbb{E}_{x\sim\mathbb{P}_n}[f^\dagger(t(x))]$. Then from Assumption 3.4 and Lemma D.6, we know that $\widehat{\mathcal{L}}(\cdot)$ is strongly convex with a constant coefficient. Note that by triangular inequality, we have

$$|\widehat{D}_f(q\|p) - D_f(q\|p)| = |\widehat{\mathcal{L}}(\widehat{t}) - \mathcal{L}(t^*)| \le |\widehat{\mathcal{L}}(t^*) - \widehat{\mathcal{L}}(\widehat{t})| + |\widehat{\mathcal{L}}(t^*) - \mathcal{L}(t^*)| =: A_1 + A_2.$$

We proceed to bound $A_1$ and $A_2$.

**Bound on $A_1$:** Recall that $\widehat{\mathcal{L}}(\cdot)$ is strongly convex. Consequently, we have

$$A_1 \lesssim \|t^* - \widehat{t}\|_{L_2(\mathbb{P})}^2 \lesssim n^{-\frac{\beta}{2\beta+d}} \log^{7/2} n,$$

with probability at least $1 - \exp\{-n^{d/(2\beta+d)} \log^5 n\}$, where the last inequality comes from (B.2).

**Bound on $A_2$:** Note that both the functions $t^*(\cdot)$ and $f^\dagger(t^*(\cdot))$ are bounded, then by Hoeffding's inequality, we obtain that

$$\mathbb{P}(A_2 \le n^{-\frac{\beta}{2\beta+d}} \log^{7/2} n) \ge 1 - \exp\{-n^{(d-2\beta)/(2\beta+d)} \log^{14} n\}.$$

Therefore, by combining the above two bounds, we obtain that

$$|\widehat{D}_f(q\|p) - D_f(q\|p)| \lesssim n^{-\frac{\beta}{2\beta+d}} \log^{7/2} n$$

with probability at least $1 - \exp\{-n^{(d-2\beta)/(2\beta+d)} \log^{14} n\}$. This concludes the proof of the theorem.

### B.4 PROOF OF THEOREM 5.3

We first need to bound the max deviation of the estimated $f$-divergence $\widehat{D}_f(q\|p)$ among all $q \in \mathcal{Q}$. The following lemma provides such a bound.

**Lemma B.1.** Under the assumptions stated in Theorem 5.3, for any fixed density $p$, if the sample size $n$ is sufficiently large, it holds that

$$\sup_{q\in\mathcal{Q}} |D_f(q\|p) - \widehat{D}_f(q\|p)| \lesssim n^{-\frac{2\beta}{2\beta+d}} \cdot \log^7 n$$

with probability at least $1 - 1/n$.

We defer the proof to Section C.1.

Now we turn to the proof of the theorem. We denote by $\widetilde{q}' = \operatorname{argmin}_{\widetilde{q}\in\mathcal{Q}} D_f(\widetilde{q}\|p)$, then with probability at least $1 - 1/n$, we have

$$\begin{aligned} D_f(\widehat{q}\|p) &\le |D_f(\widehat{q}\|p) - \widehat{D}_f(\widehat{q}\|p)| + \widehat{D}_f(\widehat{q}\|p) \\ &\le \sup_{q\in\mathcal{Q}} |D_f(q\|p) - \widehat{D}_f(q\|p)| + \widehat{D}_f(\widetilde{q}'\|p) \lesssim n^{-\frac{2\beta}{2\beta+d}} \cdot \log^7 n + D_f(\widetilde{q}'\|p). \end{aligned} \quad \text{(B.3)}$$

Here in the second line we use the optimality of $\widehat{q}$ among all $\widetilde{q} \in \mathcal{Q}$ to the problem (5.2), while the last inequality uses Lemma B.1 and Theorem 4.2. Moreover, note that $D_f(\widetilde{q}'\|p) = \min_{\widetilde{q}\in\mathcal{Q}} D_f(\widetilde{q}\|p)$, combining (B.3), it holds that with probability at least $1 - 1/n$,

$$D_f(\widehat{q}\|p) \lesssim n^{-\frac{2\beta}{2\beta+d}} \cdot \log^7 n + \min_{\widetilde{q}\in\mathcal{Q}} D_f(\widetilde{q}\|p).$$

This concludes the proof of the theorem.

### B.5 PROOF OF THEOREM A.1

For any real-valued function $\varrho$, we write $\mathbb{E}_\mathbb{P}(\varrho) = \mathbb{E}_{x\sim\mathbb{P}}[\varrho(x)]$, $\mathbb{E}_\mathbb{Q}(\varrho) = \mathbb{E}_{x\sim\mathbb{Q}}[\varrho(x)]$, $\mathbb{E}_{\mathbb{P}_n}(\varrho) = \mathbb{E}_{x\sim\mathbb{P}_n}[\varrho(x)]$, and $\mathbb{E}_{\mathbb{Q}_n}(\varrho) = \mathbb{E}_{x\sim\mathbb{Q}_n}[\varrho(x)]$ for notational convenience.

For any $\widetilde{t} \in \Phi_M(L, k, s)$, we establish the following lemma.

**Lemma B.2.** Under the assumptions stated in Theorem A.1, it holds that
$$1/(4L_0) \cdot \|\widehat{t} - \widetilde{t}\|^2_{L_2(\mathbb{P})} \leq 1/\mu_0 \cdot \|\widehat{t} - \widetilde{t}\|_{L_2(\mathbb{P})} \cdot \|\widetilde{t} - t^*\|_{L_2(\mathbb{P})} + \{\mathbb{E}_{\mathbb{Q}_n}[(\widehat{t} - \widetilde{t})/2] - \mathbb{E}_{\mathbb{Q}}[(\widehat{t} - \widetilde{t})/2]\}$$
$$- \{\mathbb{E}_{\mathbb{P}_n}[f^\dagger((\widehat{t} + \widetilde{t})/2) - f^\dagger(\widetilde{t})] - \mathbb{E}_{\mathbb{P}}[f^\dagger((\widehat{t} + \widetilde{t})/2) - f^\dagger(\widetilde{t})]\}$$
Here $\mu_0$ and $L_0$ are specified in Assumption 3.4.

We defer the proof to Section C.2.

Note that by Lemma B.2 and the fact that $f^\dagger$ is Lipschitz continuous, we have
$$\|\widehat{t} - \widetilde{t}\|^2_{L_2(\mathbb{P})} \lesssim \|\widehat{t} - \widetilde{t}\|_{L_2(\mathbb{P})} \cdot \|\widetilde{t} - t^*\|_{L_2(\mathbb{P})} + \{\mathbb{E}_{\mathbb{Q}_n}[(\widehat{t} - \widetilde{t})/2] - \mathbb{E}_{\mathbb{Q}}[(\widehat{t} - \widetilde{t})/2]\}$$
$$- \{\mathbb{E}_{\mathbb{P}_n}[f^\dagger((\widehat{t} + \widetilde{t})/2) - f^\dagger(\widetilde{t})] - \mathbb{E}_{\mathbb{P}}[f^\dagger((\widehat{t} + \widetilde{t})/2) - f^\dagger(\widetilde{t})]\}. \qquad (B.4)$$
Furthermore, to bound the RHS of the above inequality, we establish the following lemma.

**Lemma B.3.** We assume that the function $\psi : \mathbb{R} \to \mathbb{R}$ is Lipschitz continuous and bounded such that $|\psi(x)| \leq M_0$ for any $|x| \leq M$. Then under the assumptions stated in Theorem A.1, for any fixed $\widetilde{t}(x) \in \Phi_M$, $n \gtrsim [\gamma + \gamma^{-1}\log(1/\varepsilon)]^2$ and $0 < \varepsilon < 1$, we have the follows
$$\mathbb{P}\left\{\sup_{t(\cdot) \in \Phi_M(L,k,s)} \frac{|\mathbb{E}_{\mathbb{P}_n}[\psi(t) - \psi(\widetilde{t})] - \mathbb{E}_{\mathbb{P}}[\psi(t) - \psi(\widetilde{t})]|}{\eta(n,\gamma,\varepsilon) \cdot \|\psi(t) - \psi(\widetilde{t})\|_{L_2(\mathbb{P})} \vee \lambda(n,\gamma,\varepsilon)} \leq 16M_0\right\} \geq 1 - \varepsilon \cdot \exp(-\gamma^2),$$
where $\eta(n,\gamma,\varepsilon) = n^{-1/2}[\gamma\log n + \gamma^{-1}\log(1/\varepsilon)]$, $\lambda(n,\gamma,\varepsilon) = n^{-1}[\gamma^2 + \log(1/\varepsilon)]$, and for any real numbers $c_1$ and $c_2$, we denote by $c_1 \vee c_2 = \max\{c_1, c_2\}$. Here $\gamma$ takes the form $\gamma = s^{1/2}\log(V^2L)$, where $V = \prod_{j=0}^{L+1}(k_j + 1)$.

We defer the proof to Section C.3.

Note that the results in Lemma B.3 also apply to the distribution $\mathbb{Q}$, and by using the fact that the true density ratio $\theta^*(x; p, q) = q(x)/p(x)$ is bounded below and above, we know that $L_2(\mathbb{Q})$ is indeed equivalent to $L_2(\mathbb{P})$. We thus focus on $L_2(\mathbb{P})$ here. By (B.4), Lemma B.3, and the Lipschitz property of $f^\dagger$ according to Lemma D.6, with probability at least $1 - \varepsilon \cdot \exp(-\gamma^2)$, we have the following bound
$$\|\widehat{t} - \widetilde{t}\|^2_{L_2(\mathbb{P})} \lesssim \|\widehat{t} - \widetilde{t}\|_{L_2(\mathbb{P})} \cdot \|\widetilde{t} - t^*\|_{L_2(\mathbb{P})}$$
$$+ \mathcal{O}(n^{-1/2}[\gamma\log n + \gamma^{-1}\log(1/\varepsilon)] \cdot \|\widehat{t} - \widetilde{t}\|_{L_2(\mathbb{P})} \vee n^{-1}[\gamma^2 + \log(1/\varepsilon)]), \qquad (B.5)$$
where we recall that the notation $\gamma = s^{1/2}\log(V^2L)$ is a parameter related with the family of neural networks $\Phi_M$. We proceed to analyze the dominant part on the RHS of (B.5).

**Case 1.** If the term $\|\widehat{t} - \widetilde{t}\|_{L_2(\mathbb{P})} \cdot \|\widetilde{t} - t^*\|_{L_2(\mathbb{P})}$ dominates, then with probability at least $1 - \varepsilon \cdot \exp(-\gamma^2)$
$$\|\widehat{t} - \widetilde{t}\|_{L_2(\mathbb{P})} \lesssim \|\widetilde{t} - t^*\|_{L_2(\mathbb{P})}.$$

**Case 2.** If the term $\mathcal{O}(n^{-1/2}[\gamma\log n + \gamma^{-1}\log(1/\varepsilon)] \cdot \|\widehat{t} - \widetilde{t}\|_{L_2(\mathbb{P})})$ dominates, then with probability at least $1 - \varepsilon \cdot \exp(-\gamma^2)$
$$\|\widehat{t} - \widetilde{t}\|_{L_2(\mathbb{P})} \lesssim n^{-1/2}[\gamma\log n + \gamma^{-1}\log(1/\varepsilon)].$$

**Case 3.** If the term $\mathcal{O}(n^{-1}[\gamma^2 + \log(1/\varepsilon)])$ dominates, then with probability at least $1 - \varepsilon \cdot \exp(-\gamma^2)$
$$\|\widehat{t} - \widetilde{t}\|_{L_2(\mathbb{P})} \lesssim n^{-1/2}[\gamma + \sqrt{\log(1/\varepsilon)}].$$

Therefore, by combining the above three cases, we have
$$\|\widehat{t} - \widetilde{t}\|_{L_2(\mathbb{P})} \lesssim \|\widetilde{t} - t^*\|_{L_2(\mathbb{P})} + \gamma n^{-1/2}\log n + n^{-1/2}[\sqrt{\log(1/\varepsilon)} + \gamma^{-1}\log(1/\varepsilon)].$$
Further the triangular inequality gives us
$$\|\widehat{t} - t^*\|_{L_2(\mathbb{P})} \lesssim \|\widetilde{t} - t^*\|_{L_2(\mathbb{P})} + \gamma n^{-1/2}\log n + n^{-1/2}[\sqrt{\log(1/\varepsilon)} + \gamma^{-1}\log(1/\varepsilon)]$$
with probability at least $1 - \varepsilon \cdot \exp(-\gamma^2)$. Note that the above error bound holds for any $\widetilde{t} \in \Phi_M(L, k, s)$, especially for the choice $\widetilde{t}$ such that it minimizes $\|\widetilde{t} - t^*\|_{L_2(\mathbb{P})}$. Therefore, we have
$$\|\widehat{t} - t^*\|_{L_2(\mathbb{P})} \lesssim \min_{\widetilde{t} \in \Phi_M(L,k,s)} \|\widetilde{t} - t^*\|_{L_2(\mathbb{P})} + \gamma n^{-1/2}\log n + n^{-1/2}[\sqrt{\log(1/\varepsilon)} + \gamma^{-1}\log(1/\varepsilon)]$$
with probability at least $1 - \varepsilon \cdot \exp(-\gamma^2)$. This concludes the proof of the theorem.

## B.6    PROOF OF THEOREM A.3

We follow the proof in Li et al. (2018). We denote by the loss function in (A.2) as $\mathcal{L}[t(x)] = f^\dagger(t(x^\mathrm{I})) - t(x^\mathrm{II})$, where $x^\mathrm{I}$ follows the distribution $\mathbb{P}$ and $x^\mathrm{II}$ follows $\mathbb{Q}$. To prove the theorem, we first link the generalization error in our theorem to the empirical Rademacher complexity (ERC). Given the data $\{x_i\}_{i=1}^n$, the ERC related with the class $\mathcal{L}(\Phi_{\mathrm{norm}})$ is defined as

$$\mathfrak{R}_n[\mathcal{L}(\Phi_{\mathrm{norm}})] = \mathbb{E}_\varepsilon\left[\sup_{\varphi\in\Phi_{\mathrm{norm}}} |\frac{1}{n}\sum_{i=1}^n \varepsilon_i \cdot \mathcal{L}[\varphi(x_i; W, v)]|\{x_i\}_{i=1}^n\right], \tag{B.6}$$

where $\varepsilon_i$'s are i.i.d. Rademacher random variables, i.e., $\mathbb{P}(\varepsilon_i = 1) = \mathbb{P}(\varepsilon_i = -1) = 1/2$. Here the expectation $\mathbb{E}_\varepsilon(\cdot)$ is taken over the Rademacher random variables $\{\varepsilon_i\}_{i\in[n]}$.

We introduce the following Lemma B.4 (Mohri et al., 2018), which links the ERC to the generalization error bound.

**Lemma B.4.** Assume that $\sup_{\varphi\in\Phi_{\mathrm{norm}}} |\mathcal{L}(\varphi)| \leq M_1$, then for any $\varepsilon > 0$, with probability at least $1 - \varepsilon$, we have

$$\sup_{\varphi\in\Phi_{\mathrm{norm}}}\left\{\mathbb{E}_x\{\mathcal{L}[\varphi(x; W, v)]\} - \frac{1}{n}\sum_{i=1}^n \mathcal{L}[\varphi(x_i; W, v)]\right\} \lesssim \mathfrak{R}_n[\mathcal{L}(\Phi_{\mathrm{norm}})] + M_1 \cdot n^{-1/2}\sqrt{\log(1/\varepsilon)},$$

where the expectation $\mathbb{E}_x\{\cdot\}$ is taken over $x^\mathrm{I} \sim \mathbb{P}$ and $x^\mathrm{II} \sim \mathbb{Q}$.

Equipped with the above lemma, we only need to bound the ERC defined in (B.6).

**Lemma B.5.** Let $\mathcal{L}$ be a Lipschitz continuous loss function and $\Phi_{\mathrm{norm}}$ be the family of networks defined in (A.1). We assume that the input $x \in \mathbb{R}^d$ is bounded such that $\|x\|_2 \leq B$. Then it holds that

$$\mathfrak{R}_n[\mathcal{L}(\Phi_{\mathrm{norm}})] \lesssim \gamma_1 \cdot n^{-1/2}\log(\gamma_2 n),$$

where $\gamma_1$ and $\gamma_2$ are given in (A.3).

We defer the proof to Section C.4.

Now we proceed to prove the theorem. Recall that we assume that $t^* \in \Phi_{\mathrm{norm}}$. For notational convenience, we denote by

$$\widehat{H}(t) = \mathbb{E}_{x\sim\mathbb{P}_n}[f^\dagger(t(x))] - \mathbb{E}_{x\sim\mathbb{Q}_n}[t(x)], \qquad H(t) = \mathbb{E}_{x\sim\mathbb{P}}[f^\dagger(t(x))] - \mathbb{E}_{x\sim\mathbb{Q}}[t(x)].$$

Then $\mathbb{E}[\widehat{H}(t)] = H(t)$. We proceed to bound $|\widehat{D}_f(q\|p) - D_f(q\|p)| = |\widehat{H}(\widehat{t}) - H(t^*)|$. Note that if $\widehat{H}(\widehat{t}) \geq H(t^*)$, then we have

$$0 \leq \widehat{H}(\widehat{t}) - H(t^*) \leq \widehat{H}(t^*) - H(t^*), \tag{B.7}$$

where the second inequality follows from the fact that $\widehat{t}$ is the minimizer of $\widehat{H}(\cdot)$. On the other hand, if $\widehat{H}(\widehat{t}) \leq H(t^*)$, we have

$$0 \geq \widehat{H}(\widehat{t}) - H(t^*) \geq \widehat{H}(\widehat{t}) - H(\widehat{t}), \tag{B.8}$$

where the second inequality follows that fact that $t^*$ is the minimizer of $H(\cdot)$. Therefore, by (B.7), (B.8), and the fact that $\mathcal{L}(\varphi) \lesssim \prod_{j=1}^{L+1} B_j$ for any $\varphi \in \Phi_{\mathrm{norm}}$, we deduce that

$$|\widehat{H}(\widehat{t}) - H(t^*)| \leq \sup_{t\in\Phi_{\mathrm{norm}}} |\widehat{H}(t) - H(t)| \lesssim \mathfrak{R}_n[\mathcal{L}(\Phi_{\mathrm{norm}})] + \prod_{j=1}^{L+1} B_j \cdot n^{-1/2}\sqrt{\log(1/\varepsilon)} \tag{B.9}$$

with probability at least $1 - \varepsilon$. Here the second inequality follows from Lemma B.4. By plugging the result from Lemma B.5 into (B.9), we deduce that with probability at least $1 - \varepsilon$, it holds that

$$|\widehat{D}_f(q\|p) - D_f(q\|p)| = |\widehat{H}(\widehat{t}) - H(t^*)| \lesssim \gamma_1 \cdot n^{-1/2}\log(\gamma_2 n) + \prod_{j=1}^{L+1} B_j \cdot n^{-1/2}\sqrt{\log(1/\varepsilon)}.$$

This concludes the proof of the theorem.

### B.7 Proof of Theorem A.4

We first need to bound the max deviation of the estimated $f$-divergence $\widehat{D}_f(q\|p)$ among all $q \in \mathcal{Q}$. We utilize the following lemma to provide such a bound.

**Lemma B.6.** Assume that the distribution $q$ is in the set $\mathcal{Q}$, and we denote its $L_2$ covering number as $N_2(\delta, \mathcal{Q})$. Then for any target distribution $p$, we have

$$\max_{q \in \mathcal{Q}} |D_f(q\|p) - \widehat{D}_f(q\|p)| \lesssim b_2(n, \gamma_1, \gamma_2) + \prod_{j=1}^{L+1} B_j \cdot n^{-1/2} \cdot \sqrt{\log(N_2[b_2(n, \gamma_1, \gamma_2), \mathcal{Q}]/\varepsilon)}$$

with probability at least $1 - \varepsilon$. Here $b_2(n, \gamma_1, \gamma_2) = \gamma_1 n^{-1/2} \log(\gamma_2 n)$ and $c$ is a positive absolute constant.

We defer the proof to Section C.5.

Now we turn to the proof of the theorem. We denote by $\widetilde{q}' = \operatorname{argmin}_{\widetilde{q} \in \mathcal{Q}} D_f(\widetilde{q}\|p)$. Then with probability at least $1 - \varepsilon$, we have

$$\begin{aligned}
D_f(\widehat{q}\|p) &\leq |D_f(\widehat{q}\|p) - \widehat{D}_f(\widehat{q}\|p)| + \widehat{D}_f(\widehat{q}\|p) \\
&\leq \max_{q \in \mathcal{Q}} |D_f(q\|p) - \widehat{D}_f(q\|p)| + \widehat{D}_f(\widetilde{q}'\|p) \\
&\lesssim b_2(n, \gamma_1, \gamma_2) + \prod_{j=1}^{L+1} B_j \cdot n^{-1/2} \cdot \sqrt{\log(N_2[b_2(n, \gamma_1, \gamma_2), \mathcal{Q}]/\varepsilon)} + D_f(\widetilde{q}'\|p),
\end{aligned}$$

where we use the optimality of $\widehat{q}$ among all $\widetilde{q} \in \mathcal{Q}$ to the problem (5.2) in the second inequality, and we uses Lemma B.6 and Theorem 4.2 in the last line. Moreover, note that $D_f(\widetilde{q}'\|p) = \min_{\widetilde{q} \in \mathcal{Q}} D_f(\widetilde{q}\|p)$, we obtain that

$$D_f(\widehat{q}\|p) \lesssim b_2(n, \gamma_1, \gamma_2) + \prod_{j=1}^{L+1} B_j \cdot n^{-1/2} \sqrt{\log(N_2[b_2(n, \gamma_1, \gamma_2), \mathcal{Q}]/\varepsilon)} + \min_{\widetilde{q} \in \mathcal{Q}} D_f(\widetilde{q}\|p).$$

This concludes the proof of the theorem.

## C Lemmas and Proofs

### C.1 Proof of Lemma B.1

Recall that the covering number of $\mathcal{Q}$ is $N_2(\delta, \mathcal{Q})$, we thus assume that there exists $q_1, \ldots, q_{N_2(\delta, \mathcal{Q})} \in \mathcal{Q}$ such that for any $q \in \mathcal{Q}$, there exists some $q_k$, where $1 \leq k \leq N_2(\delta, \mathcal{Q})$, so that $\|q - q_k\|_2 \leq \delta$. Moreover, by taking $\delta = \delta_n = n^{-2\beta/(2\beta+d)}$ and union bound, we have

$$\begin{aligned}
\mathbb{P}[\sup_{q \in \mathcal{Q}} &|D_f(q\|p) - \widehat{D}_f(q\|p)| \geq c_1 \cdot n^{-\frac{2\beta}{2\beta+d}} \cdot \log^7 n] \\
&\leq \sum_{k=1}^{N_2(\delta_n, \mathcal{Q})} \mathbb{P}[|D_f(q_k\|p) - \widehat{D}_f(q_k\|p)| \geq c_1 \cdot n^{-\frac{2\beta}{2\beta+d}} \cdot \log^7 n] \\
&\leq N_2(\delta_n, \mathcal{Q}) \cdot \exp(-n^{\frac{d-2\beta}{2\beta+d}} \cdot \log^{14} n),
\end{aligned}$$

where the last line comes from Theorem 4.2. Combining Assumption 5.2, when $n$ is sufficiently large, it holds that

$$\mathbb{P}[\sup_{q \in \mathcal{Q}} |D_f(q\|p) - \widehat{D}_f(q\|p)| \geq c_1 \cdot n^{-\frac{2\beta}{2\beta+d}} \cdot \log^7 n] \leq 1/n,$$

which concludes the proof of the lemma.

## C.2 Proof of Lemma B.2

For any real-valued function $\varrho$, we write $\mathbb{E}_{\mathbb{P}}(\varrho) = \mathbb{E}_{x \sim \mathbb{P}}[\varrho(x)]$, $\mathbb{E}_{\mathbb{Q}}(\varrho) = \mathbb{E}_{x \sim \mathbb{Q}}[\varrho(x)]$, $\mathbb{E}_{\mathbb{P}_n}(\varrho) = \mathbb{E}_{x \sim \mathbb{P}_n}[\varrho(x)]$, and $\mathbb{E}_{\mathbb{Q}_n}(\varrho) = \mathbb{E}_{x \sim \mathbb{Q}_n}[\varrho(x)]$ for notational convenience.

By the definition of $\widehat{t}$ in (4.3), we have

$$\mathbb{E}_{\mathbb{P}_n}[f^\dagger(\widehat{t})] - \mathbb{E}_{\mathbb{Q}_n}(\widehat{t}) \leq \mathbb{E}_{\mathbb{P}_n}[f^\dagger(\widetilde{t})] - \mathbb{E}_{\mathbb{Q}_n}(\widetilde{t}).$$

Note that the functional $G(t) = \mathbb{E}_{\mathbb{P}_n}[f^\dagger(t)] - \mathbb{E}_{\mathbb{Q}_n}(t)$ is convex in $t$ since $f^\dagger$ is convex, we then have

$$G(\frac{\widehat{t} + \widetilde{t}}{2}) - G(\widetilde{t}) \leq \frac{G(\widehat{t}) - G(\widetilde{t})}{2} \leq 0.$$

By re-arranging terms, we have

$$\{\mathbb{E}_{\mathbb{P}_n}[f^\dagger((\widehat{t}+\widetilde{t})/2) - f^\dagger(\widetilde{t})] - \mathbb{E}_{\mathbb{P}}[f^\dagger((\widehat{t}+\widetilde{t})/2) - f^\dagger(\widetilde{t})]\} - \{\mathbb{E}_{\mathbb{Q}_n}[(\widehat{t}-\widetilde{t})/2] - \mathbb{E}_{\mathbb{Q}}[(\widehat{t}-\widetilde{t})/2]\}$$
$$\leq \mathbb{E}_{\mathbb{Q}}[(\widehat{t}-\widetilde{t})/2] - \mathbb{E}_{\mathbb{P}}[f^\dagger((\widehat{t}+\widetilde{t})/2) - f^\dagger(\widetilde{t})]. \tag{C.1}$$

We denote by

$$B_f(\widetilde{t}, t) = \mathbb{E}_{\mathbb{P}}[f^\dagger(t) - f^\dagger(\widetilde{t})] - \mathbb{E}_{\mathbb{Q}}(t - \widetilde{t}). \tag{C.2}$$

then the RHS of (C.1) is exactly $-B_f(\widetilde{t}, (\widehat{t}+\widetilde{t})/2)$. We proceed to establish the lower bound of $B_f(\widetilde{t}, t)$ using $L_2(\mathbb{P})$ norm. From $t^*(x; p, q) = f'(q(x)/p(x))$ and $(f^\dagger)' \circ (f')(x) = x$, we know that $q/p = \partial f^\dagger(t^*)/\partial t$. Then by substituting the second term on the RHS of (C.2) using the above relationship, we have

$$B_f(\widetilde{t}, t) = \mathbb{E}_{\mathbb{P}}\left[f^\dagger(t) - f^\dagger(\widetilde{t}) - \frac{\partial f^\dagger}{\partial t}(t^*) \cdot (t - \widetilde{t})\right]$$
$$= \mathbb{E}_{\mathbb{P}}\left[f^\dagger(t) - f^\dagger(\widetilde{t}) - \frac{\partial f^\dagger}{\partial t}(\widetilde{t}) \cdot (t - \widetilde{t})\right] + \mathbb{E}_{\mathbb{P}}\left\{\left[\frac{\partial f^\dagger}{\partial t}(\widetilde{t}) - \frac{\partial f^\dagger}{\partial t}(t^*)\right] \cdot (t - \widetilde{t})\right\}$$
$$= A_1 + A_2. \tag{C.3}$$

We lower bound $A_1$ and $A_2$ in the sequel.

**Bound on $A_1$.** Note that by Assumption 3.4 and Lemma D.6, we know that the Fenchel duality $f^\dagger$ is strongly convex with parameter $1/L_0$. This gives that

$$f^\dagger(t(x)) - f^\dagger(\widetilde{t}(x)) - \frac{\partial f^\dagger}{\partial t}(\widetilde{t}(x)) \cdot [t(x) - \widetilde{t}(x)] \geq 1/L_0 \cdot (t(x) - \widetilde{t}(x))^2$$

for any $x$. Consequently, it holds that

$$A_1 \geq 1/L_0 \cdot \|t - \widetilde{t}\|^2_{L_2(\mathbb{P})}. \tag{C.4}$$

**Bound on $A_2$.** By Cauchy-Schwarz inequality, it holds that

$$A_2 \geq -\sqrt{\mathbb{E}_{\mathbb{P}}\left\{\left[\frac{\partial f^\dagger}{\partial t}(\widetilde{t}) - \frac{\partial f^\dagger}{\partial t}(t^*)\right]^2\right\}} \cdot \sqrt{\mathbb{E}_{\mathbb{P}}[(t - \widetilde{t})^2]}.$$

Again, by Assumption 3.4 and Lemma D.6, we know that the Fenchel duality $f^\dagger$ has $1/\mu_0$-Lipschitz gradient, which gives that

$$\left|\frac{\partial f^\dagger}{\partial t}(\widetilde{t}(x)) - \frac{\partial f^\dagger}{\partial t}(t^*(x))\right| \leq 1/\mu_0 \cdot |\widetilde{t}(x) - t^*(x)|$$

for any $x$. By this, the term $A_2$ is lower bounded:

$$A_2 \geq -1/\mu_0 \cdot \|\widetilde{t} - t^*\|_{L_2(\mathbb{P})} \cdot \|t - \widetilde{t}\|_{L_2(\mathbb{P})}. \tag{C.5}$$

Plugging (C.4) and (C.5) into (C.3), we have

$$B_f(\widetilde{t}, t) \geq 1/L_0 \cdot \|t - \widetilde{t}\|^2_{L_2(\mathbb{P})} - 1/\mu_0 \cdot \|\widetilde{t} - t^*\|_{L_2(\mathbb{P})} \cdot \|t - \widetilde{t}\|_{L_2(\mathbb{P})}.$$

By this, together with (C.1), we conclude that

$$1/(4L_0) \cdot \|\widehat{t} - \widetilde{t}\|^2_{L_2(\mathbb{P})} \leq 1/\mu_0 \cdot \|\widehat{t} - \widetilde{t}\|_{L_2(\mathbb{P})} \cdot \|\widetilde{t} - t^*\|_{L_2(\mathbb{P})} + \{\mathbb{E}_{\mathbb{Q}_n}[(\widehat{t}-\widetilde{t})/2] - \mathbb{E}_{\mathbb{Q}}[(\widehat{t}-\widetilde{t})/2]\}$$
$$- \{\mathbb{E}_{\mathbb{P}_n}[f^\dagger((\widehat{t}+\widetilde{t})/2) - f^\dagger(\widetilde{t})] - \mathbb{E}_{\mathbb{P}}[f^\dagger((\widehat{t}+\widetilde{t})/2) - f^\dagger(\widetilde{t})]\}.$$

This concludes the proof of the lemma.

## C.3 Proof of Lemma B.3

For any real-valued function $\varrho$, we write $\mathbb{E}_{\mathbb{P}}(\varrho) = \mathbb{E}_{x \sim \mathbb{P}}[\varrho(x)]$, $\mathbb{E}_{\mathbb{Q}}(\varrho) = \mathbb{E}_{x \sim \mathbb{Q}}[\varrho(x)]$, $\mathbb{E}_{\mathbb{P}_n}(\varrho) = \mathbb{E}_{x \sim \mathbb{P}_n}[\varrho(x)]$, and $\mathbb{E}_{\mathbb{Q}_n}(\varrho) = \mathbb{E}_{x \sim \mathbb{Q}_n}[\varrho(x)]$ for notational convenience.

We first introduce the following concepts. For any $K > 0$, the Bernstein difference $\rho_{K,\mathbb{P}}^2(t)$ of $t(\cdot)$ with respect to the distribution $\mathbb{P}$ is defined to be

$$\rho_{K,\mathbb{P}}^2(t) = 2K^2 \cdot \mathbb{E}_{\mathbb{P}}[\exp(|t|/K) - 1 - |t|/K].$$

Correspondingly, we denote by $\mathcal{H}_{K,B}$ the generalized entropy with bracketing induced by the Bernstein difference $\rho_{K,\mathbb{P}}$. We denote by $H_{s,B}$ the entropy with bracketing induced by $L_s$ norm, $H_s$ the entropy induced by $L_s$ norm, $H_{L_s(\mathbb{P}),B}$ the entropy with bracketing induced by $L_s(\mathbb{P})$ norm, and $H_{L_s(\mathbb{P})}$ the regular entropy induced by $L_s(\mathbb{P})$ norm.

Since we focus on fixed $L$, $k$, and $s$, we denote by $\Phi_M = \Phi_M(L, k, s)$ for notational convenience. We consider the space

$$\Psi_M = \psi(\Phi_M) = \{\psi(t) : t(x) \in \Phi_M\}.$$

For any $\delta > 0$, we denote the following space

$$\Psi_M(\delta) = \{\psi(t) \in \Psi_M : \|\psi(t) - \psi(\widetilde{t})\|_{L_2(\mathbb{P})} \le \delta\},$$
$$\Psi_M'(\delta) = \{\Delta\psi(t) = \psi(t) - \psi(\widetilde{t}) : \psi(t) \in \Psi_M(\delta)\}.$$

Note that $\sup_{\Delta\psi(t) \in \Psi_M'(\delta)} \|\Delta\psi(t)\|_\infty \le 2M_0$ and $\sup_{\Delta\psi(t) \in \Psi_M'(\delta)} \|\Delta\psi(t)\|_\infty \le \delta$, by Lemma D.4 we have

$$\sup_{\Delta\psi(t) \in \Psi_M'(\delta)} \rho_{8M_0,\mathbb{P}}[\Delta\psi(t)] \le \sqrt{2}\delta.$$

To invoke Theorem D.3 for $\mathcal{G} = \Psi_M'(\delta)$, we pick $K = 8M_0$ and $R = \sqrt{2}\delta$. Note that from the fact that $\sup_{\Delta\psi(t) \in \Psi_M'(\delta)} \|\Delta\psi(t)\|_\infty \le 2M_0$, by Lemma D.1, Lemma D.2, and the fact that $\psi$ is Lipschitz continuous, we have

$$\mathcal{H}_{8M_0,B}(u, \Psi_M'(\delta), \mathbb{P}) \le H_\infty(u/(2\sqrt{2}), \Psi_M'(\delta)) \le 2(s+1)\log(4\sqrt{2}u^{-1}(L+1)V^2)$$

for any $u > 0$. Then, by algebra, we have the follows

$$\int_0^R \mathcal{H}_{8M_0,B}^{1/2}(u, \Psi_M'(\delta), \mathbb{P})\,\mathrm{d}u \le 3s^{1/2}\delta \cdot \log(8V^2L/\delta).$$

For any $0 < \varepsilon < 1$, we take $C = 1$, and $a, C_1$ and $C_0$ in Theorem D.3 to be

$$a = 8M_0 \log(\exp(\gamma^2)/\varepsilon)\gamma^{-1} \cdot \delta,$$
$$C_0 = 6M_0\gamma^{-1}\sqrt{\log(\exp(\gamma^2)/\varepsilon)},$$
$$C_1 = 33M_0^2\gamma^{-2}\log(\exp(\gamma^2)/\varepsilon).$$

Here $\gamma = s^{1/2}\log(V^2L)$. Then it is straightforward to check that our choice above satisfies the conditions in Theorem D.3 for any $\delta$ such that $\delta \ge \gamma n^{-1/2}$, when $n$ is sufficiently large such that $n \gtrsim [\gamma + \gamma^{-1}\log(1/\varepsilon)]^2$. Consequently, by Theorem D.3, for $\delta \ge \gamma n^{-1/2}$, we have

$$\mathbb{P}\{\sup_{t(x) \in \Phi_M(\delta)} |\mathbb{E}_{\mathbb{P}_n}[\psi(t) - \psi(\widetilde{t})] - \mathbb{E}_{\mathbb{P}}[\psi(t) - \psi(\widetilde{t})]| \ge 8M_0 \log(\exp(\gamma^2)/\varepsilon)\gamma^{-1} \cdot \delta \cdot n^{-1/2}\}$$

$$= \mathbb{P}\{\sup_{\Delta\psi(t) \in \Psi_M'(\delta)} |\mathbb{E}_{\mathbb{P}_n}[\Delta\psi(t)] - \mathbb{E}_{\mathbb{P}}[\Delta\psi(t)]| \ge 8M_0 \log(\exp(\gamma^2)/\varepsilon)\gamma^{-1} \cdot \delta \cdot n^{-1/2}\}$$

$$\le \varepsilon \cdot \exp(-\gamma^2).$$

By taking $\delta = \delta_n = \gamma n^{-1/2}$, we have

$$\mathbb{P}\left\{\sup_{t(x) \in \Phi_M(\delta)} \frac{|\mathbb{E}_{\mathbb{P}_n}[\psi(t) - \psi(\widetilde{t})] - \mathbb{E}_{\mathbb{P}}[\psi(t) - \psi(\widetilde{t})]|}{n^{-1}[\gamma^2 + \log(1/\varepsilon)]} \le 8M_0\right\} \ge 1 - \varepsilon \cdot \exp(-\gamma^2). \quad \text{(C.6)}$$

On the other hand, we denote that $S = \min\{s > 1 : 2^{-s}(2M_0) < \delta_n\} = \mathcal{O}(\log(\gamma^{-1}n^{1/2}))$. For notational convenience, we denote the set

$$A_s = \{\psi(t) \in \Psi_M : \psi(t) \in \Psi_M(2^{-s+2}M_0), \psi(t) \notin \Psi_M(2^{-s+1}M_0)\}. \tag{C.7}$$

Then by the peeling device, we have the following

$$\mathbb{P}\left\{\sup_{\psi(t)\in\Psi_M,\psi(t)\notin\Psi_M(\delta_n)} \frac{|\mathbb{E}_{\mathbb{P}_n}[\psi(t)-\psi(\widetilde{t})] - \mathbb{E}_{\mathbb{P}}[\psi(t)-\psi(\widetilde{t})]|}{\|\psi(t)-\psi(\widetilde{t})\|_{L_2(\mathbb{P})} \cdot T(n,\gamma,\varepsilon)} \geq 16M_0\right\}$$

$$\leq \sum_{s=1}^{S} \mathbb{P}\left\{\sup_{\psi(t)\in A_s} \frac{|\mathbb{E}_{\mathbb{P}_n}[\psi(t)-\psi(\widetilde{t})] - \mathbb{E}_{\mathbb{P}}[\psi(t)-\psi(\widetilde{t})]|}{2^{-s+1}M_0} \geq 16M_0 \cdot T(n,\gamma,\varepsilon)\right\}$$

$$\leq \sum_{s=1}^{S} \mathbb{P}\{\sup_{\psi(t)\in A_s} |\mathbb{E}_{\mathbb{P}_n}[\psi(t)-\psi(\widetilde{t})] - \mathbb{E}_{\mathbb{P}}[\psi(t)-\psi(\widetilde{t})]| \geq 8M_0 \cdot (2^{-s+2}M_0) \cdot T(n,\gamma,\varepsilon)\}$$

$$\leq \sum_{s=1}^{S} \mathbb{P}\{\sup_{\psi(t)\in\Psi_M(2^{-s+2}M_0)} |\mathbb{E}_{\mathbb{P}_n}[\psi(t)-\psi(\widetilde{t})] - \mathbb{E}_{\mathbb{P}}[\psi(t)-\psi(\widetilde{t})]| \geq 8M_0 \cdot (2^{-s+2}M_0) \cdot T(n,\gamma,\varepsilon)\}$$

$$\leq S \cdot \varepsilon \cdot \exp(-\gamma^2)/\log(\gamma^{-1}n^{1/2}) = c \cdot \varepsilon \cdot \exp(-\gamma^2),$$

where $c$ is a positive absolute constant, and for notational convenience we denote by $T(n,\gamma,\varepsilon) = \gamma^{-1} \cdot n^{-1/2}\log(\log(\gamma^{-1}n^{1/2})\exp(\gamma^2)/\varepsilon)$. Here in the second line, we use the fact that for any $\psi(t) \in A_s$, we have $\|\psi(t)-\psi(\widetilde{t})\|_{L_2(\mathbb{Q})} \geq 2^{-s+1}M_0$ by the definition of $A_s$ in (C.7); in the forth line, we use the argument that since $A_s \subseteq \Psi_M(2^{-s+2}M_0)$, the probability of supremum taken over $\Psi_M(2^{-s+2}M_0)$ is larger than the one over $A_s$; in the last line we invoke Theorem D.3. Consequently, this gives us

$$\mathbb{P}\left\{\sup_{\substack{\psi(t)\in\Psi_M \\ \psi(t)\notin\Psi_M(\delta_n)}} \frac{|\mathbb{E}_{\mathbb{P}_n}[\psi(t)-\psi(\widetilde{t})] - \mathbb{E}_{\mathbb{P}}[\psi(t)-\psi(\widetilde{t})]|}{\|\psi(t)-\psi(\widetilde{t})\|_{L_2(\mathbb{P})} \cdot n^{-1/2}[\gamma\log n + \gamma^{-1}\log(1/\varepsilon)]} \leq 16M_0\right\} \geq 1 - \varepsilon \cdot \exp(-\gamma^2).$$
$$\tag{C.8}$$

Combining (C.6) and (C.8), we finish the proof of the lemma.

## C.4 PROOF OF LEMMA B.5

The proof of the theorem utilizes following two lemmas. The first lemma characterizes the Lipschitz property of $\varphi(x; W, v)$ in the input $x$.

**Lemma C.1.** Given $W$ and $v$, then for any $\varphi(\cdot; W, v) \in \Phi_{\text{norm}}$ and $x_1, x_2 \in \mathbb{R}^d$, we have

$$\|\varphi(x_1; W, v) - \varphi(x_2; W, v)\|_2 \leq \|x_1 - x_2\|_2 \cdot \prod_{j=1}^{L+1} B_j.$$

We defer the proof to Section C.6.

The following lemma characterizes the Lipschitz property of $\varphi(x; W, v)$ in the network parameter pair $(W, v)$.

**Lemma C.2.** Given any bounded $x \in \mathbb{R}^d$ such that $\|x\|_2 \leq B$, then for any weights $W^1 = \{W_j^1\}_{j=1}^{L+1}, W^2 = \{W_j^2\}_{j=1}^{L+1}, v^1 = \{v_j^1\}_{j=1}^{L}, v^2 = \{v_j^2\}_{j=1}^{L}$, and functions $\varphi(\cdot, W^1, v^1), \varphi(\cdot, W^2, v^2) \in \Phi_{\text{norm}}$, we have

$$\|\varphi(x, W^1, v^1) - \varphi(x, W^2, v^2)\|$$
$$\leq \frac{B\sqrt{2L+1} \cdot \prod_{j=1}^{L+1} B_j}{\min_j B_j} \cdot \sum_{j=1}^{L} A_j \cdot \sqrt{\sum_{j=1}^{L+1} \|W_j^1 - W_j^2\|_F^2 + \sum_{j=1}^{L} \|v_j^1 - v_j^2\|_2^2}.$$

We defer the proof to Section C.7.

We now turn to the proof of Lemma B.5. Note that by Lemma C.2, we know that $\varphi(x; W, v)$ is $L_w$-Lipschitz in the parameter $(W, v) \in \mathbb{R}^b$, where the dimension $b$ takes the form

$$b = \sum_{j=1}^{L+1} k_j k_{j-1} + \sum_{j=1}^{L} k_j \leq \sum_{j=0}^{L+1} (k_j + 1)^2, \tag{C.9}$$

and the Lipschitz constant $L_w$ satisfies

$$L_w = \frac{B\sqrt{2L+1} \cdot \prod_{j=1}^{L+1} B_j}{\min_j B_j} \cdot \sum_{j=1}^{L} A_j. \tag{C.10}$$

In addition, we know that the covering number of $\mathcal{W} = \{(W, v) \in \mathbb{R}^b : \sum_{j=1}^{L+1} \|W_j\|_F + \sum_{j=1}^{L} \|v_j\|_2 \leq K\}$, where

$$K = \sqrt{\sum_{j=1}^{L+1} k_j^2 B_j^2 + \sum_{j=1}^{L} A_j}, \tag{C.11}$$

satisfies

$$N(\mathcal{W}, \delta) \leq (3K\delta^{-1})^b.$$

By the above facts, we deduce that the covering number of $\mathcal{L}(\Phi_{\text{norm}})$ satisfies

$$N[\mathcal{L}(\Phi_{\text{norm}}), \delta] \leq (c_1 K L_w \delta^{-1})^b,$$

for some positive absolute constant $c_1$. Then by Dudley entropy integral bound on the ERC, we know that

$$\mathfrak{R}_n[\mathcal{L}(\Phi_{\text{norm}})] \leq \inf_{\tau > 0} \tau + \frac{1}{\sqrt{n}} \int_\tau^\vartheta \sqrt{\log N[\mathcal{L}(\Phi_{\text{norm}}), \delta]} \, d\delta, \tag{C.12}$$

where $\vartheta = \sup_{g(\cdot; W, v) \in \mathcal{L}(\Phi_{\text{norm}}), x \in \mathbb{R}^d} |g(x; W, v)|$. Moreover, from Lemma C.1 and the fact that the loss function is Lipschitz continuous, we have

$$\vartheta \leq c_2 \cdot B \cdot \prod_{j=1}^{L+1} B_j \tag{C.13}$$

for some positive absolute constant $c_2$. Therefore, by calculations, we derive from (C.12) that

$$\mathfrak{R}_n[\mathcal{L}(\Phi_{\text{norm}})] = \mathcal{O}\left(\frac{\vartheta}{\sqrt{n}} \cdot \sqrt{b \cdot \log \frac{K L_w \sqrt{n}}{\vartheta \sqrt{b}}}\right),$$

then we conclude the proof of the lemma by plugging in (C.9), (C.10), (C.11), and (C.13), and using the definition of $\gamma_1$ and $\gamma_2$ in (A.3).

## C.5  PROOF OF LEMMA B.6

Remember that the covering number of $\mathcal{Q}$ is $N_2(\delta, \mathcal{Q})$, we assume that there exists $q_1, \ldots, q_{N_2(\delta, \mathcal{Q})} \in \mathcal{Q}$ such that for any $q \in \mathcal{Q}$, there exists some $q_k$, where $1 \leq k \leq N_2(\delta, \mathcal{Q})$, so that $\|q - q_k\|_2 \leq \delta$. Moreover, by taking $\delta = \gamma_1 n^{-1/2} \log(\gamma_2 n) = b_2(n, \gamma_1, \gamma_2)$ and $N_2 = N_2[b_2(n, \gamma_1, \gamma_2), \mathcal{Q}]$, we have

$$\mathbb{P}\{\max_{q \in \mathcal{Q}} |D_f(q\|p) - \widehat{D}_f(q\|p)| \geq c \cdot [b_2(n, \gamma_1, \gamma_2) + \prod_{j=1}^{L+1} B_j \cdot n^{-1/2} \cdot \sqrt{\log(N_2/\varepsilon)}]\}$$

$$\leq \sum_{k=1}^{N_2} \mathbb{P}\{|D_f(q\|p) - \widehat{D}_f(q\|p)| \geq c \cdot [b_2(n, \gamma_1, \gamma_2) + \prod_{j=1}^{L+1} B_j \cdot n^{-1/2} \cdot \sqrt{\log(N_2/\varepsilon)}]\}$$

$$\leq N_2 \cdot \varepsilon/N_2 = \varepsilon,$$

where the second line comes from union bound, and the last line comes from Theorem A.3. By this, we conclude the proof of the lemma.

## C.6  PROOF OF LEMMA C.1

The proof follows by applying the Lipschitz property and bounded spectral norm of $W_j$ recursively:

$$
\begin{aligned}
\|\varphi(x_1; W, v) - \varphi(x_2; W, v)\|_2 &= \|W_{L+1}(\sigma_{v_L} \cdots W_2 \sigma_{v_1} W_1 x_1 - \sigma_{v_L} \cdots W_2 \sigma_{v_1} W_1 x_2)\|_2 \\
&\leq \|W_{L+1}\|_2 \cdot \|\sigma_{v_L}(W_L \cdots W_2 \sigma_{v_1} W_1 x_1 - W_L \cdots W_2 \sigma_{v_1} W_1 x_2)\|_2 \\
&\leq B_{L+1} \cdot \|W_L \cdots W_2 \sigma_{v_1} W_1 x_1 - W_L \cdots W_2 \sigma_{v_1} W_1 x_2\|_2 \\
&\leq \cdots \leq \prod_{j=1}^{L+1} B_j \cdot \|x_1 - x_2\|_2.
\end{aligned}
$$

Here in the third line we uses the fact that $\|W_j\|_2 \leq B_j$ and the 1-Lipschitz property of $\sigma_{v_j}(\cdot)$, and in the last line we recursively apply the same argument as in the above lines. This concludes the proof of the lemma.

## C.7  PROOF OF LEMMA C.2

Recall that $\varphi(x; W, v)$ takes the form

$$
\varphi(x; W, v) = W_{L+1} \sigma_{v_L} W_L \cdots \sigma_{v_1} W_1 x.
$$

For notational convenience, we denote by $\varphi_j^i(x) = \sigma_{v_j^i}(W_j^i x)$ for $i = 1, 2$. By this, $\varphi(x; W, v)$ has the form $\varphi(x; W^i, v^i) = W_{L+1}^i \varphi_L^i \circ \cdots \circ \varphi_1^i(x)$. First, note that for any $W^1, W^2, v^1$ and $v^2$, by triangular inequality, we have

$$
\begin{aligned}
\|\varphi(x, W^1, v^1) - \varphi(x, W^2, v^2)\|_2 &= \|W_{L+1}^1 \varphi_L^1 \circ \cdots \circ \varphi_1^1(x) - W_{L+1}^2 \varphi_L^2 \circ \cdots \circ \varphi_1^2(x)\|_2 \\
&\leq \|W_{L+1}^1 \varphi_L^1 \circ \cdots \circ \varphi_1^1(x) - W_{L+1}^2 \varphi_L^1 \circ \cdots \circ \varphi_1^1(x)\|_2 \\
&\quad + \|W_{L+1}^2 \varphi_L^1 \circ \cdots \circ \varphi_1^1(x) - W_{L+1}^2 \varphi_L^2 \circ \cdots \circ \varphi_1^2(x)\|_2 \\
&\leq \|W_{L+1}^1 - W_{L+1}^2\|_F \cdot \|\varphi_L^1 \circ \cdots \circ \varphi_1^1(x)\|_2 \\
&\quad + B_{L+1} \cdot \|\varphi_L^1 \circ \cdots \circ \varphi_1^1(x) - \varphi_L^2 \circ \cdots \circ \varphi_1^2(x)\|_2. \quad (C.14)
\end{aligned}
$$

Moreover, note that for any $\ell \in [L]$, we have the following bound on $\|\varphi_L^1 \circ \cdots \circ \varphi_1^1(x)\|_2$:

$$
\begin{aligned}
\|\varphi_\ell^i \circ \cdots \circ \varphi_1^i(x)\|_2 &\leq \|W_\ell^i \varphi_{\ell-1}^i \circ \cdots \circ \varphi_1^i(x)\|_2 + \|v_\ell^i\|_2 \\
&\leq B_\ell \cdot \|\varphi_{\ell-1}^i \circ \cdots \circ \varphi_1^i(x)\|_2 + A_\ell \\
&\leq \|x\|_2 \cdot \prod_{j=1}^\ell B_j + \sum_{j=1}^\ell A_j \prod_{i=j+1}^\ell B_i, \quad (C.15)
\end{aligned}
$$

where the first inequality comes from the triangle inequality, and the second inequality comes from the bounded spectral norm of $W_j^i$, while the last inequality simply applies the previous arguments recursively. Therefore, combining (C.14), we have

$$
\begin{aligned}
\|\varphi(x, W^1, v^1) - \varphi(x, W^2, v^2)\|_2 &\leq \left( B \cdot \prod_{j=1}^L B_j + \sum_{j=1}^L A_j \prod_{i=j+1}^L B_i \right) \cdot \|W_{L+1}^1 - W_{L+1}^2\|_F \\
&\quad + B_{L+1} \cdot \|\varphi_L^1 \circ \cdots \circ \varphi_1^1(x) - \varphi_L^2 \circ \cdots \circ \varphi_1^2(x)\|_2. \quad (C.16)
\end{aligned}
$$

Similarly, by triangular inequality, we have

$$
\begin{aligned}
\|\varphi_L^1 \circ \cdots \circ \varphi_1^1(x) - \varphi_L^2 \circ \cdots \circ \varphi_1^2(x)\|_2 \\
\leq \|\varphi_L^1 \circ \varphi_{L-1}^1 \circ \cdots \circ \varphi_1^1(x) - \varphi_L^2 \circ \varphi_{L-1}^1 \circ \cdots \circ \varphi_1^1(x)\|_2 \\
\quad + \|\varphi_L^2 \circ \varphi_{L-1}^1 \circ \cdots \circ \varphi_1^1(x) - \varphi_L^2 \circ \varphi_{L-1}^2 \circ \cdots \circ \varphi_1^2(x)\|_2 \\
\leq \|\varphi_L^1 \circ \varphi_{L-1}^1 \circ \cdots \circ \varphi_1^1(x) - \varphi_L^2 \circ \varphi_{L-1}^1 \circ \cdots \circ \varphi_1^1(x)\|_2 \quad (C.17) \\
\quad + B_L \cdot \|\varphi_{L-1}^1 \circ \cdots \circ \varphi_1^1(x) - \varphi_{L-1}^2 \circ \cdots \circ \varphi_1^2(x)\|_2,
\end{aligned}
$$

where the second inequality uses the bounded spectral norm of $W_L$ and 1-Lipschitz property of $\sigma_{v_L}(\cdot)$. For notational convenience, we further denote $y = \varphi_{L-1}^1 \circ \cdots \circ \varphi_1^1(x)$, then

$$\|\varphi_L^1(y) - \varphi_L^2(y)\|_2 = \|\sigma(W_L^1 y - v_L^1) - \sigma(W_L^2 y - v_L^2)\}\|_2$$
$$\leq \|v_L^1 - v_L^2\|_2 + \|W_L^1 - W_L^2\|_F \cdot \|y\|_2,$$

where the inequality comes from the 1-Lipschitz property of $\sigma(\cdot)$. Moreover, combining (C.15), it holds that

$$\|\varphi_L^1(y) - \varphi_L^2(y)\|_2 \leq \|v_L^1 - v_L^2\|_2 + \|W_L^1 - W_L^2\|_F \cdot \left( B \cdot \prod_{j=1}^{L-1} B_j + \sum_{j=1}^{L-1} A_j \prod_{i=j+1}^{L-1} B_i \right). \quad \text{(C.18)}$$

By (C.17) and (C.18), we have

$$\|\varphi_L^1 \circ \cdots \circ \varphi_1^1(x) - \varphi_L^2 \circ \cdots \circ \varphi_1^2(x)\|_2$$

$$\leq \|v_L^1 - v_L^2\|_2 + \|W_L^1 - W_L^2\|_F \cdot \left( B \cdot \prod_{j=1}^{L-1} B_j + \sum_{j=1}^{L-1} A_j \prod_{i=j+1}^{L-1} B_i \right)$$

$$\quad + B_L \cdot \|\varphi_{L-1}^1 \circ \cdots \circ \varphi_1^1(x) - \varphi_{L-1}^2 \circ \cdots \circ \varphi_1^2(x)\|_2$$

$$\leq \sum_{j=1}^{L} \prod_{i=j+1}^{L} B_i \cdot \|v_j^1 - v_j^2\|_2 + \frac{B \cdot \prod_{j=1}^{L+1} B_j}{\min_j B_j} \cdot \sum_{j=1}^{L} A_j \cdot \sum_{j=1}^{L} \|W_j^1 - W_j^2\|_F$$

$$\leq \frac{B \cdot \prod_{j=1}^{L+1} B_j}{\min_j B_j} \cdot \sum_{j=1}^{L} A_j \cdot \sum_{j=1}^{L} (\|v_j^1 - v_j^2\|_2 + \|W_j^1 - W_j^2\|_F).$$

Here in the second inequality we recursively apply the previous arguments. Further combining (C.16), we obtain that

$$\|\varphi(x, W^1, v^1) - \varphi(x, W^2, v^2)\|_2$$

$$\leq \frac{B \cdot \prod_{j=1}^{L+1} B_j}{\min_j B_j} \cdot \sum_{j=1}^{L} A_j \cdot \left( \sum_{j=1}^{L+1} \|W_j^1 - W_j^2\|_F + \sum_{j=1}^{L} \|v_j^1 - v_j^2\|_2 \right)$$

$$\leq \frac{B\sqrt{2L+1} \cdot \prod_{j=1}^{L+1} B_j}{\min_j B_j} \cdot \sum_{j=1}^{L} A_j \cdot \sqrt{\sum_{j=1}^{L+1} \|W_j^1 - W_j^2\|_F^2 + \sum_{j=1}^{L} \|v_j^1 - v_j^2\|_2^2},$$

where we use Cauchy-Schwarz inequality in the last line. This concludes the proof of the lemma.

## D  AUXILIARY RESULTS

**Lemma D.1.** The following statements for entropy hold.

1. Suppose that $\sup_{g \in \mathcal{G}} \|g\|_\infty \leq M$, then

$$\mathcal{H}_{4M,B}(\sqrt{2}\delta, \mathcal{G}, \mathbb{Q}) \leq H_{2,B}(\delta, \mathcal{G}, \mathbb{Q})$$

for any $\delta > 0$.

2. For $1 \leq q < \infty$, and $\mathbb{Q}$ a distribution, we have

$$H_{p,B}(\delta, \mathcal{G}, \mathbb{Q}) \leq H_\infty(\delta/2, \mathcal{G}),$$

for any $\delta > 0$. Here $H_\infty$ is the entropy induced by infinity norm.

3. Based on the above two statements, suppose that $\sup_{g \in \mathcal{G}} \|g\|_\infty \leq M$, we have

$$\mathcal{H}_{4M,B}(\sqrt{2} \cdot \delta, \mathcal{G}, \mathbb{Q}) \leq H_\infty(\delta/2, \mathcal{G}),$$

by taking $p = 2$.

*Proof.* See van de Geer & van de Geer (2000) for a detailed proof. □

**Lemma D.2.** The entropy of the neural network set defined in (4.1) satisfies

$$H_\infty[\delta, \Phi_M(L, p, s)] \leq (s+1)\log(2\delta^{-1}(L+1)V^2),$$

where $V = \prod_{l=0}^{L+1}(p_l + 1)$.

*Proof.* See Schmidt-Hieber (2017) for a detailed proof. □

**Theorem D.3.** Assume that $\sup_{g\in\mathcal{G}} \rho_K(g) \leq R$. Take $a$, $C$, $C_0$, and $C_1$ satisfying that $a \leq C_1\sqrt{n}R^2/K$, $a \leq 8\sqrt{n}R$, $a \geq C_0 \cdot [\int_0^R H_{K,B}^{1/2}(u, \mathcal{G}, \mathbb{P})du \vee R]$, and $C_0^2 \geq C^2(C_1 + 1)$. It holds that

$$\mathbb{P}[\sup_{g\in\mathcal{G}} |\mathbb{E}_{\mathbb{P}_n}(g) - \mathbb{E}_{\mathbb{P}}(g)| \geq a \cdot n^{-1/2}] \leq C\exp\left(-\frac{a^2}{C^2(C_1+1)R^2}\right).$$

*Proof.* See van de Geer & van de Geer (2000) for a detailed proof. □

**Lemma D.4.** Suppose that $\|g\|_\infty \leq K$, and $\|g\| \leq R$, then $\rho_{2K,\mathbb{P}}^2(g) \leq 2R^2$. Moreover, for any $K' \geq K$, we have $\rho_{2K',\mathbb{P}}^2(g) \leq 2R^2$.

*Proof.* See van de Geer & van de Geer (2000) for a detailed proof. □

**Theorem D.5.** For any function $f$ in the Hölder ball $\mathcal{C}_d^\beta([0,1]^d, K)$ and any integers $m \geq 1$ and $N \geq (\beta+1)^d \vee (K+1)$, there exists a network $\widetilde{f} \in \Phi(L, (d, 12dN, \ldots, 12dN, 1), s)$ with number of layers $L = 8 + (m+5)(1 + \lceil\log_2 d\rceil)$ and number of parameters $s \leq 94d^2(\beta+1)^{2d}N(m+6)(1 + \lceil\log_2 d\rceil)$, such that

$$\|\widetilde{f} - f\|_{L^\infty([0,1]^d)} \leq (2K+1)3^{d+1}N2^{-m} + K2^\beta N^{-\beta/d}.$$

*Proof.* See Schmidt-Hieber (2017) for a detailed proof. □

**Lemma D.6.** If the function $f$ is strongly convex with parameter $\mu_0 > 0$ and has Lipschitz continuous gradient with parameter $L_0 > 0$, then the Fenchel duality $f^\dagger$ of $f$ is $1/L_0$-strongly convex and has $1/\mu_0$-Lipschitz continuous gradient (therefore, $f^\dagger$ itself is Lipschitz continuous).

*Proof.* See Zhou (2018) for a detailed proof. □

