# OpenReview forum: "Credible Sample Elicitation by Deep Learning, for Deep Learning"
_ICLR.cc/2020/Conference — Reject_

### Official Review · AnonReviewer2 · 2019-10-22
**Official Blind Review #2**

**Rating:** 6

**Review:**

This paper proposes a sample elicitation framework to tackle the problem of eliciting credible samples from agents for complex distributions. The authors suggest that deep neural frameworks can be applied in this framework for sample elicitation through the derivations. The authors also show the connection between the problem of sample elicitation and f-GAN. However, some problems in the proof on sample elicitation should be clarified or carefully explained.

- In (C.15) in Section C.7, why the upper bound is irrelevant to y or W_L^1 or W_L^2? Since we do not have W_L^1 = W_L^2, how to derive the inequality in (C.15)?

- In the proof of Lemma B.2 in Section B.5, why 1/(4L_0) and (1/\mu_0) are simply removed in the inequality below?

- The inequalities in the proof of Theorem 3.5 in Section B.1 should be further explained. What is the meaning of “agent believes p = q” and how to apply this to the lower bound? What is the meaning of \bar{S} and how to get these inequalities for the upper bound?

Some minor comments
- The relation of x’ and x in the first paragraph on page 2 should be further clarified, and some formal definition should be given. The readers who are not familiar with this area would be confused about the problem before reading Section 2.1.


**Experience Assessment:**

I do not know much about this area.

**Review Assessment: Checking Correctness Of Derivations And Theory:**

I carefully checked the derivations and theory.

**Review Assessment: Checking Correctness Of Experiments:**

N/A

**Review Assessment: Thoroughness In Paper Reading:**

N/A

---

> ### Author Response · Authors · 2019-11-06
> **thank you for the catches & clarifications**
>
> Yes, the upper bound should be relevant to y, W_L^1, and W_L^2.  We are sorry about the typo, but it does not affect the results much by slight modification.  In specific, to obtain the accurate result as stated in Theorem A.3, we only need to multiply by \sum_{j=1}^L A_j in the definition of \gamma_2 in (A.3) (with Theorem A.3 being unchanged).
>
> In the proof of Lemma B.2 in Section B.5, the “less than or equal to” sign should be changed to “asymptotically less than or equal to” sign. In this way, the inequality holds since both 1/(4L_0) and 1/\mu_0 are constants.
>
> Again, we apologize for these typos, and we thank the reviewer for pointing them out for the improvement of our paper.  We have already uploaded a revision to reflect these changes.
>
> Theorem 3.5: “p=q” is a typical argument in information elicitation.  In order to induce truthful reporting, the agent needs to reason about the expectation of his score w.r.t. His true belief. And the agents are assumed to believe that his *truthful* samples come from the same distribution as the ground truth ones. The lower bound is due to the following facts: i) the $(\delta(n), \epsilon(n))$ estimation guarantee of the divergence function informs us $\hat{D}_f(q||p) \leq D_f(q||p)+\epsilon(n)$ (with probability at least $1-\delta(n)$), the true value plus an error term. ii) then p=q allows us to change $D_f(q||p)$ to $D_f(p||p)$.
>
> $\bar{S}$ is the max of $S$ - but here is a typo, we do not need this $\bar{S}$ term. The upper bounds holds as $D_f(p||\tilde{p}) \geq 0$ by definition of divergence. Plug in we derived the claim. We are updating the draft to reflect the catch.
>
> We will clarify the relationships between a report x and a reference sample x’ in the introduction.
>
> We hope this helps clarify the confusions! Again thank you for all the catches.

---

### Official Review · AnonReviewer1 · 2019-10-24
**Official Blind Review #1**

**Rating:** 1

**Review:**

- Summary

This paper studies the sample elicitation problem where agents are asked to report samples. The goal is then to evaluate the quality of these reported samples by means of a scoring function S. Following previous related works, the authors use the equivalence between maximizing the expected proper score and minimizing some f-divergence. Their approach relies on the dual expression of the f-divergence which writes as a maximum over a set of functions t. Theoretical guarantees are given for f-scorings obtained (with or without ground truth samples) by first computing the empirical optimal function t, then plugged to estimate the f-divergence. Finally, a deep learning approach is proposed by considering functions f parameterized as sparse deep neural networks.

- Critics

The paper is globally well written but not well motivated and sometimes difficult to understand.
In particular, the notions of "elicitation", "reports" and "score function" should be defined mathematically more clearly.
Moreover, the deep learning aspect of the paper is not well motivated and is introduced in a very arbitrary way. Why not choosing another parametric family of functions? Is there another (broad) family of functions for which the computation of the argmin in Equation (4.3) is more tractable in practice?
A convincing way to motivate this deep learning approach would be to include numerical experiments and to compare to other parametric families.


**Experience Assessment:**

I have read many papers in this area.

**Review Assessment: Checking Correctness Of Derivations And Theory:**

I carefully checked the derivations and theory.

**Review Assessment: Checking Correctness Of Experiments:**

I carefully checked the experiments.

**Review Assessment: Thoroughness In Paper Reading:**

I read the paper thoroughly.

---

> ### Author Response · Authors · 2019-11-06
> **clarification of our motivation, and why deep learning**
>
> Since ICLR is the premier deep learning conference, we are motivated to collect credible and quality samples from strategic agents (e.g., ordinary people) for deep learning. Naturally, we think it is of interest to try using deep learning techniques to solve the score function design problem via a data-driven approach. Along the way of developing our results, we realized the connection between our elicitation problem and GAN, which we detail in Section 5.
>
> Beyond above, the deep learning based estimators are able to handle complex data. And with our deep learning solution, we are further able to provide estimates for the divergence functions used for our scoring mechanisms, with provable finite sample complexity. In our opinion, these are highly non-trivial contributions. In this paper, we focus on developing theoretical guarantees- other parametric families either can not handle complex data, e.g., it is hard to handle images using kernel methods, or do not have provable guarantees on the sample complexity.
>
> We wonder whether there is another reason that leads to the reviewer’s recommendation of rejection. We are happy to further clarify.

---

### Author Response · Authors · 2019-11-12
**Revision uploaded**

Dear reviewers,

We have updated our draft according to your comments and uploaded a revised draft. We have better motivated why we consider using deep learning to design a data-driven elicitation mechanism. We have also fixed typos and made several clarifications. We thank all reviewers for the helpful comments.

Best,
Authors

---

### Decision · Program_Chairs · 2019-12-19

**Decision:**

Reject

**Comment:**

The primary contribution of this manuscript is a conceptual and theoretical solution to the sample elicitation problem, where agents are asked to report samples. The procedure is implemented using score functions to evaluate the quality of the samples.

The reviewers and AC agree that the problem studied is timely and interesting, as there is limited work on credible sample elicitation in the literature. However, the reviewers were unconvinced about the motivation of the work, and the clarity of the conceptual results. There is also a lack of empirical evaluation. IN the opinion of the AC, this manuscript, while interesting, can be improved by significant revision for clarity and context, and revisions should ideally include some empirical evaluation.